# Rethinking Moreau Envelope for Nonconvex Bi-Level Optimization: A Single-loop and Hessian-free Solution Strategy

## Abstract

Bi-Level Optimization (BLO) has found diverse applications in machine learning due to its ability to model nested structures. Addressing large-scale BLO problems for complex learning tasks presents two significant challenges: ensuring computational efficiency and providing theoretical guarantees. Recent advancements in scalable BLO algorithms has predominantly relied on lower-level convexity simplification. In this context, our work takes on the challenge of large-scale BLO problems involving nonconvexity in both the upper and lower levels. We address both computational and theoretical challenges simultaneously. Specifically, by utilizing the Moreau envelope-based reformulation, we introduce an innovative single-loop gradient-based algorithm with non-asymptotic convergence analysis for general nonconvex BLO problems. Notably, this algorithm relies solely on first-order gradient information, making it exceedingly practical and efficient, particularly for large-scale BLO learning tasks. We validate the effectiveness of our approach on a series of different synthetic problems, two typical hyper-parameter learning tasks and the real-world neural architecture search application. These experiments collectively substantiate the superior performance of our method.

## 1 Introduction

Bi-Level Optimization (BLO) addresses the challenges posed by nested optimization structures that arise in a wide range of machine learning applications, such as hyper-parameter optimization Pedregosa (2016); Franceschi et al. (2018); Mackay et al. (2019), meta learning Zügner & Günnemann (2018); Rajeswaran et al. (2019); Ji et al. (2020), neural architecture search Liu et al. (2018); Chen et al. (2019); Elsken et al. (2020), etc. Refer to recent survey papers Liu et al. (2021a); Zhang et al. (2023) for more applications of BLO in machine learning, computer vision and signal processing.

The inherent nested nature gives rise to several difficulties and hurdles in effectively solving BLO problems. Over the past decade, a large number of BLO methods have emerged, with a primary emphasis on addressing BLO problems featuring strongly convex lower-level (LL) objective. The LL strong convexity assumption ensures the uniqueness of LL minimizer (a.k.a., LL Singleton), which simplifies both the optimization process and theoretical analysis, see, e.g., Franceschi et al. (2018); Grazzi et al. (2020); Ghadimi & Wang (2018); Hong et al. (2023); Chen et al. (2021); Ji et al. (2021; 2022). To mitigate the restrictive LL Singleton condition, another line of research is dedicated to BLO with convex LL problems, which bring about several challenges such as the presence of multiple LL optimal solutions (a.k.a., Non-Singleton). This may hinder the adoption of implicit-based approaches that rely on the implicit function theorem. To address this concern, recent advances include: aggregation methods Liu et al. (2020); Li et al. (2020); Liu et al. (2022; 2023b); difference-of-convex algorithm Gao et al. (2022); Ye et al. (2023); primal-dual algorithms Sow et al. (2022a); first-order penalty methods Lu & Mei (2023).

While the nonconvex-convex BLO has been extensively studied in the literature, efficient methods for nonconvex-nonconvex BLO remain under-explored. Beyond LL convexity, Liu et al. (2021c) proposed an iterative differentiation-based BLO method; Arbel & Mairal (2022b) extended the approximate implicit differentiation approach; Liu et al. (2021b) utilized the value function reformulation of BLO to develop algorithms in machine learning. All of these works, however, tend to

Table 1: Comparison of our method MEHA with closely related works for addressing **nonconvex-nonconvex BLO** ( IAPTT-GM Liu et al. (2021c), BOME! Ye et al. (2022), V-PBGD Shen & Chen (2023), GALET Xiao et al. (2023) ). Different methods employ distinct stationary measures, so we do not delve into complexity comparison here. Below, PL Condition represents the Polyak-Łojasiewicz (PL) condition; $L$-Smooth means the Lipschitz continuous gradient condition; Bounded and Gradient-Bounded specify that $|F(x,y)| \leq C$ and $\|\nabla_y F(x,y)\| \leq C$ for all $(x,y)$, respectively; $F$ and $f$ are UL and LL objectives, respectively.

| Method | Upper-Level Objective | Lower-Level Objective | Hessian Free | Single Loop | Non-Asymptotic |
|---|---|---|---|---|---|
| IAPTT-GM | Smooth | $L$-Smooth & Compactness | ✗ | ✗ | ✗ |
| GALET | $L$-Smooth & Gradient-Bounded | PL Condition & $L$-Smooth | ✗ | ✗ | ✔ |
| BOME! | $L$-Smooth & Bounded & Gradient-Bounded | PL Condition & $L$-Smooth | ✔ | ✗ | ✔ |
| V-PBGD | $L$-Smooth & Gradient-Bounded | PL Condition & $L$-Smooth | ✔ | ✗ | ✔ |
| MEHA (smooth case) | $L$-Smooth | $L$-Smooth | ✔ | ✔ | ✔ |
| MEHA (general case) | $L$-Smooth | $L$-Smooth Part + Weakly Convex Nonsmooth Part | ✔ | ✔ | ✔ |

be complicated and impractical for large-scale BLO problems, and lack a non-asymptotic analysis. When LL objective satisfies the Polyak-Łojasiewicz (PL) or local PL conditions, Ye et al. (2022) introduced a fully first-order value function-based BLO algorithm with non-asymptotic convergence analysis. Recently, while still considering the PL condition, Huang (2023) introduced a momentum-based BLO algorithm; Xiao et al. (2023) proposed a generalized alternating method; Shen & Chen (2023) proposed a penalty-based fully first-order BLO algorithm. However, the existing methods still present two significant challenges: ensuring computational efficiency and offering theoretical guarantees in the absence of the PL condition. A summary of the comparison of the proposed method with closely related works is provided in Table 1.

## 1.1 MAIN CONTRIBUTIONS

To the best of our knowledge, this work is the first study to utilize Moreau envelope-based reformulation of BLO, originally presented in Gao et al. (2023), to design a single-loop and Hessian-free gradient-based algorithm with non-asymptotic convergence analysis for general BLO problems with potentially nonconvex and nonsmooth LL objective functions. This setting encompasses a wide range of machine learning applications, see, e.g., the recent surveys Liu et al. (2021a); Zhang et al. (2023). Conducting non-asymptotic analysis for our algorithm, which addresses nonconvex LL problem, poses substantial challenges. Existing single-loop gradient-based methods generally require the LL objective to either be strongly convex or satisfy the PL condition, as a mechanism to control the approximation errors incurred when utilizing a single gradient descent step to approximate the real LL optimal solution. Our approach mitigates this limitation by employing Moreau envelope-based reformulation, where the proximal LL problem may exhibit strong convexity even if the original LL problem is nonconvex. Consequently, this enables effective error control and facilitates the algorithm's non-asymptotic convergence analysis for nonconvex LL problem. We summarize our contributions as follows.

- We propose the Moreau Envelope reformulation based Hessian-free Algorithm (MEHA), for general BLO problems with nonconvex and probably nonsmooth LL objective functions. MEHA avoids second-order derivative approximations related to the Hessian matrix and can be implemented efficiently in a single-loop manner, enhancing its practicality and efficiency for large-scale nonconvex-nonconvex BLO in deep learning.

- We provide a rigorous analysis of the non-asymptotic convergence of MEHA under milder conditions, avoiding the need for either the convexity assumption or the PL condition on LL problem. In the context of the smooth BLO scenario, our assumption simplifies to UL and LL objective functions being $L$-smooth.

- We validate the effectiveness and efficiency of MEHA on various synthetic problems, two typical hyper-parameter learning tasks and the real-world neural architecture search application. These experiments collectively substantiate the superior performance of MEHA.

## 1.2 RELATED WORK

We give a brief review of some recent works that are directly related to ours. An expanded review of recent studies is provided in Section A.1. Beyond LL convexity, the recent work Huang (2023) proposes a momentum-based implicit gradient BLO algorithm and establishes a convergence analysis framework under the PL condition and some nondegenerate condition of LL Hessian. The study Xiao et al. (2023) introduces a novel stationary metric for nonconvex-PL BLOs and develops a generalized alternating method under the PL Condition. Due to the use of implicit gradient or the KKT reformulation, these methods require computationally intensive operations related to the Hessian matrix. On the other hand, Ye et al. (2022) presents a Hessian-free (also known as fully first-order) BLO algorithm utilizing the value function reformulation, which comes with non-asymptotic convergence guarantees under the PL or local PL conditions. However, it involves a double-loop structure. Recently, Shen & Chen (2023) develops a penalty-based fully first-order algorithm for both unconstrained and constrained BLOs. They establish its finite-time convergence under the PL conditions, although it employs a double-loop structure.

## 2 A SINGLE-LOOP AND HESSIAN-FREE SOLUTION STRATEGY

In this work, we study a bi-level optimization (BLO) problem:

$$\min_{x \in X, y \in Y} F(x, y) \quad \text{s.t.} \quad y \in S(x), \tag{1}$$

where $S(x)$ denotes the set of optimal solutions for the lower-level (LL) problem given by

$$\min_{y \in Y} \varphi(x, y) := f(x, y) + g(x, y), \tag{2}$$

where $X$ and $Y$ are closed convex sets in $\mathbb{R}^n$ and $\mathbb{R}^m$, respectively. The function $f(x, y) : \mathbb{R}^n \times \mathbb{R}^m \to \mathbb{R}$ is smooth, and generally nonconvex, while $g(x, y) : \mathbb{R}^n \times \mathbb{R}^m \to \mathbb{R}$ is potentially nonsmooth with respect to (w.r.t.) the LL variable $y$. For specific conditions governing $F$, $f$ and $g$, we refer the reader to Assumptions 3.1- 3.2.

## 2.1 MOREAU ENVELOPE BASED REFORMULATION

In this study, we do not necessitate convexity assumptions on the LL problem. Drawing inspiration from the Moreau envelope based reformulation introduced in Gao et al. (2023) for convex LL scenarios, we define the Moreau envelope $v_\gamma(x, y)$ associated with the LL problem as follows:

$$v_\gamma(x, y) := \inf_{\theta \in Y} \left\{ \varphi(x, \theta) + \frac{1}{2\gamma} \|\theta - y\|^2 \right\}, \tag{3}$$

where $\gamma > 0$. By leveraging this Moreau envelope, we investigate a reformulated version of the original BLO problem,

$$\min_{(x,y) \in X \times Y} F(x, y) \quad \text{s.t.} \quad \varphi(x, y) - v_\gamma(x, y) \leq 0. \tag{4}$$

It should be noted that $\varphi(x, y) \geq v_\gamma(x, y)$ holds for all $(x, y) \in X \times Y$. For the special case where $\varphi(x, y)$ is convex in $y \in Y$ for any $x \in X$, the equivalence between the reformulated and the original BLO problems is established in (Gao et al., 2023, Theorem 2.1). In the absence of convexity assumptions on $\varphi$, but when $\varphi(x, \cdot)$ is $\rho_{\varphi_2}$-weakly convex [1] on $Y$ and $\gamma \in (0, 1/\rho_{\varphi_2})$, we establish

---

[1] A function $h : \mathbb{R}^p \to \mathbb{R} \cup \{\infty\}$ is $\rho$-weakly convex if $h(z) + \frac{\rho}{2}\|z\|^2$ is convex. In the context where $z = (x, y)$, we always say $h$ is $(\rho_1, \rho_2)$-weakly convex if $h(x, y) + \frac{\rho_1}{2}\|x\|^2 + \frac{\rho_2}{2}\|y\|^2$ is convex.

in Theorem A.1 that the reformulation (4) is equivalent to a relaxed version of BLO problem (1),

$$\min_{x \in X, y \in Y} F(x, y) \quad \text{s.t.} \quad y \in \tilde{S}(x) := \{y \mid 0 \in \nabla_y f(x, y) + \partial_y g(x, y) + \mathcal{N}_Y(y)\}, \qquad (5)$$

where $\partial_y g(x, y)$ denotes the partial Fréchet (regular) subdifferential of $g$ w.r.t. the LL variable at $(x, y)$, and $\mathcal{N}_Y(y)$ signifies the normal cone to $Y$ at $y$. The stationary condition characterizing $\tilde{S}(x)$ is the optimality conditions of the lower-level problem within the setting of this work, specifically, Assumption 3.2. This can be validated through the application of subdifferential sum rules, see, e.g., (Mordukhovich, 2018, Proposition 1.30, Theorem 2.19).

Specifically, the reformulated problem (4) becomes equivalent to the original BLO problem when the set $\tilde{S}(x)$ coincides with $S(x)$. This equivalence holds, for instances, when $\varphi(x, \cdot)$ is convex or $\varphi(x, y) \equiv f(x, y)$ and it satisfies the PL condition, that is, there exists $\mu > 0$ such that for any $x \in X$, the inequality $\|\nabla_y f(x, y)\|^2 \geq 2\mu \big(f(x, y) - \inf_{\theta \in \mathbb{R}^m} f(x, \theta)\big)$ holds for all $y \in \mathbb{R}^m$.

Before presenting our proposed method, we briefly review some relevant preliminary results related to $v_\gamma(x, y)$, with a special focus on its gradient properties. Assuming that $\varphi(x, y)$ is $(\rho_{\varphi_1}, \rho_{\varphi_2})$-weakly convex on $X \times Y$, we demonstrate that for $\gamma \in (0, \frac{1}{2\rho_{\varphi_2}})$, the function $v_\gamma(x, y) + \frac{\rho_{v_1}}{2}\|x\|^2 + \frac{\rho_{v_2}}{2}\|y\|^2$ is convex over $X \times \mathbb{R}^m$ when $\rho_{v_1} \geq \rho_{\varphi_1}$ and $\rho_{v_2} \geq 1/\gamma$. This result implies that $v_\gamma(x, y)$ is weakly convex, as detailed in Lemma A.1. Lastly, we define

$$S_\gamma(x, y) := \operatorname{argmin}_{\theta \in Y} \left\{ \varphi(x, \theta) + \frac{1}{2\gamma}\|\theta - y\|^2 \right\}. \qquad (6)$$

For $\gamma \in (0, \frac{1}{2\rho_{\varphi_2}})$, the solution set $S_\gamma(x, y) = \{\theta_\gamma^*(x, y)\}$ is a singleton. Further, when the gradients $\nabla_x f(x, y)$ and $\nabla_x g(x, y)$ exist, the gradient of $v_\gamma(x, y)$ can be expressed as follows,

$$\nabla v_\gamma(x, y) = \Big( \nabla_x f(x, \theta_\gamma^*(x, y)) + \nabla_x g(x, \theta_\gamma^*(x, y)), \ \big(y - \theta_\gamma^*(x, y)\big)/\gamma \Big), \qquad (7)$$

which is established in Lemma A.2.

## 2.2 SINGLE-LOOP MOREAU ENVELOPE BASED HESSIAN-FREE ALGORITHM (MEHA)

We introduce a single-loop algorithm for the general BLO problem (1), via solving the reformulated problem (4). At each iteration, using the current iterate $(x^k, y^k, \theta^k)$, we update the variable $\theta$ by conducting a single proximal gradient iteration on the proximal LL problem (3), as follows,

$$\theta^{k+1} = \operatorname{Prox}_{\eta_k \tilde{g}(x^k, \cdot)} \left( \theta^k - \eta_k \left( \nabla_y f(x^k, \theta^k) + \frac{1}{\gamma}(\theta^k - y^k) \right) \right), \qquad (8)$$

where $\eta_k$ is the stepsize, and $\tilde{g}(x, y) := g(x, y) + \delta_Y(y)$ represents the nonsmooth part of the LL problem. Here, we define $\operatorname{Prox}_h(y)$ as the proximal mapping of a function $h : \mathbb{R}^m \to \mathbb{R} \cup \{\infty\}$,

$$\operatorname{Prox}_h(y) := \arg \min_{\theta \in \mathbb{R}^m} \left\{ h(\theta) + \|\theta - y\|^2/2 \right\}.$$

Subsequently, we update the variables $(x, y)$ using the following scheme,

$$\begin{aligned} x^{k+1} &= \operatorname{Proj}_X \left( x^k - \alpha_k d_x^k \right), \\ y^{k+1} &= \operatorname{Prox}_{\beta_k \tilde{g}(x^{k+1}, \cdot)} \left( y^k - \beta_k d_y^k \right), \end{aligned} \qquad (9)$$

where $\operatorname{Proj}_X$ denotes the Euclidean projection operator, and the directions $d_x^k, d_y^k$ are defined as:

$$d_x^k := \frac{1}{c_k} \nabla_x F(x^k, y^k) + \nabla_x f(x^k, y^k) + \nabla_x g(x^k, y^k) - \nabla_x f(x^k, \theta^{k+1}) - \nabla_x g(x^k, \theta^{k+1}),$$

$$d_y^k := \frac{1}{c_k} \nabla_y F(x^{k+1}, y^k) + \nabla_y f(x^{k+1}, y^k) - \frac{1}{\gamma}(y^k - \theta^{k+1}).$$

(10)

Leveraging the formula (7), $(\nabla_x f(x^k, \theta^{k+1}) + \nabla_x g(x^k, \theta^{k+1}), (y^k - \theta^{k+1})/\gamma)$ can be regarded as an approximation for the gradient of $v_\gamma(x, y)$, with $\theta^{k+1}$ serving as a proxy for $\theta_\gamma^*(x^k, y^k)$. Hence,

the update scheme of variables $(x, y)$ in (9) can be construed as an inexact alternating proximal gradient method, operating on the following nonsmooth problem,

$$\min_{(x,y)\in X\times Y} \frac{1}{c_k} F(x, y) + f(x, y) + g(x, y) - v_\gamma(x, y).$$

Here $c_k > 0$ is a penalty parameter. The complete algorithm is outlined in Algorithm 1, whose specific form for smooth BLOs, i.e., $g(x, y) \equiv 0$, is provided in Algorithm 2 in Appendix.

---

**Algorithm 1** Single-loop Moreau Envelope based Hessian-free Algorithm (MEHA)

---

**Initialize:** $x^0, y^0, \theta^0$, stepsizes $\alpha_k, \beta_k, \eta_k$, proximal parameter $\gamma$, penalty parameter $c_k$;

1: **for** $k = 0, 1, \ldots, K - 1$ **do**

2:  update $\theta^{k+1} = \text{Prox}_{\eta_k \tilde{g}(x^k, \cdot)} \left( \theta^k - \eta_k \left( \nabla_y f(x^k, \theta^k) + \frac{1}{\gamma}(\theta^k - y^k) \right) \right)$;

3:  calculate $d_x^k, d_y^k$ as in equation 10;

4:  update
$$x^{k+1} = \text{Proj}_X \left( x^k - \alpha_k d_x^k \right),$$
$$y^{k+1} = \text{Prox}_{\beta_k \tilde{g}(x^{k+1}, \cdot)} \left( y^k - \beta_k d_y^k \right).$$

5: **end for**

---

## 3 THEORETICAL INVESTIGATIONS

### 3.1 GENERAL ASSUMPTIONS

Throughout this work, we assume the following standing assumptions on $F$, $f$ and $g$ hold.

**Assumption 3.1** (**Upper-Level Objective**). *The UL objective $F$ is bounded below on $X \times Y$, denoted by $\underline{F} := \inf_{(x,y)\in X\times Y} F(x, y) > -\infty$. Furthermore, $F$ is $L_F$-smooth[2] on $X \times Y$.*

**Assumption 3.2** (**Lower-Level Objective**). *Assume that the following conditions hold:*

(i) *The smooth component $f(x, y)$ is $L_f$-smooth on $X \times Y$.*

(ii) *The nonsmooth component $g(x, y)$ is $(\rho_{g_1}, \rho_{g_2})$-weakly convex on $X \times Y$, i.e., $g(x, y) + \frac{\rho_{g_1}}{2}\|x\|^2 + \frac{\rho_{g_2}}{2}\|y\|^2$ is convex on $X \times Y$. Additionally, the gradient $\nabla_x g(x, y)$ exists and is $L_g$-Lipschitz continuous on $X \times Y$. Moreover, let $\tilde{g}(x, y) := g(x, y) + \delta_Y(y)$, there exist positive constants $L_{\tilde{g}}, \bar{s}$ such that for any $x, x' \in X$, $\theta \in Y$ and $s \in (0, \bar{s}]$,*

$$\left\| \text{Prox}_{s\tilde{g}(x, \cdot)}(\theta) - \text{Prox}_{s\tilde{g}(x', \cdot)}(\theta) \right\| \leq L_{\tilde{g}} \|x - x'\|. \tag{11}$$

These assumptions considerably alleviate the LL problem's smoothness requirements prevalent in the BLO literature. Even within the context of smooth BLO, our assumptions only require that the UL and LL objective functions are both $L$-smooth, without imposing any conditions on the boundedness of $\nabla_y F(x, y)$, as illustrated in Table 1. Consequently, our problem setting encompasses a broad range of practical scenarios, see, e.g., the learning models in Grazzi et al. (2020).

It is noteworthy that when the nonsmooth component of the LL objective is decoupled from the UL variable, that is, $g(x, y) = \hat{g}(y)$, the condition (11) in Assumption 3.2(ii) is satisfied trivially. Consequently, Assumption 3.2(ii) holds for conventional convex regularizer functions, such as $g(x, y) = \lambda\|y\|_1$, and $g(x, y) = \lambda\|y\|_2$, where $\lambda > 0$. The demand for weak convexity is relatively lenient; a broad spectrum of functions meet this requirement. This encompasses conventional nonconvex regularizers like the Smoothly Clipped Absolute Deviation (SCAD) and the Minimax Concave Penalty (MCP) (refer to (Böhm & Wright, 2021, Section 2.1)).

Additionally, in Section A.9 of the Appendix, we show that $g(x, y) = x\|y\|_1$ fulfills Assumption 3.2(ii) when $X = \mathbb{R}_+$ and $Y = \mathbb{R}^m$.

---

[2] A function $h$ is said to be $L$-smooth on $X \times Y$ if $h$ is continuously differentiable and its gradient $\nabla h$ is $L$-Lipschitz continuous on $X \times Y$.

Finally, leveraging the descent lemma (Beck, 2017, Lemma 5.7), it can be obtained that any function featuring a Lipschitz-continuous gradient is inherently weakly convex. Thus, under Assumption 3.2(i), $f(x, y)$ is $(\rho_{f_1}, \rho_{f_2})$-weakly convex over $X \times Y$, with $\rho_{f_1} = \rho_{f_2} = L_f$. As a result, under Assumption 3.2, the LL objective function $\varphi(x, y)$ is $(\rho_{\varphi_1}, \rho_{\varphi_2})$-weakly convex on $X \times Y$, where $\rho_{\varphi_1} = \rho_{f_1} + \rho_{g_1}$ and $\rho_{\varphi_2} = \rho_{f_2} + \rho_{g_2}$.

## 3.2 CONVERGENCE RESULTS

To establish the convergence results, we first illustrate the decreasing property of the merit function:

$$V_k := \phi_{c_k}(x^k, y^k) + \left[(L_f + L_g)^2 + 1/\gamma^2\right] \left\|\theta^k - \theta_\gamma^*(x^k, y^k)\right\|^2,$$

where $\theta_\gamma^*(x, y)$ is the unique solution to problem (6) and

$$\phi_{c_k}(x, y) := \frac{1}{c_k}\left(F(x, y) - \underline{F}\right) + f(x, y) + g(x, y) - v_\gamma(x, y). \tag{12}$$

**Lemma 3.1.** *Under Assumptions 3.1 and 3.2, suppose $\gamma \in (0, \frac{1}{2\rho_{f_2} + 2\rho_{g_2}})$, $c_{k+1} \geq c_k$ and $\eta_k \in [\underline{\eta}, (1/\gamma - \rho_{f_2})/(L_f + 1/\gamma)^2) \cap [\underline{\eta}, 1/\rho_{g_2})$ with $\underline{\eta} > 0$, then there exists $c_\alpha, c_\beta, c_\theta > 0$ such that when $\alpha_k \in (0, c_\alpha]$ and $\beta_k \in (0, c_\beta]$, the sequence of $(x^k, y^k, \theta^k)$ generated by Algorithm 1 satisfies*

$$V_{k+1} - V_k \leq -\frac{1}{4\alpha_k}\|x^{k+1} - x^k\|^2 - \frac{1}{4\beta_k}\|y^{k+1} - y^k\|^2 - c_\theta\left\|\theta^k - \theta_\gamma^*(x^k, y^k)\right\|^2. \tag{13}$$

It is worth noting that $V_k$ is nonnegative for all $k$. Utilizing Lemma 3.1, we derive

$$\sum_{k=0}^\infty \frac{1}{4\alpha_k}\|x^{k+1} - x^k\|^2 + \frac{1}{4\beta_k}\|y^{k+1} - y^k\|^2 + c_\theta\left\|\theta^k - \theta_\gamma^*(x^k, y^k)\right\|^2 < \infty. \tag{14}$$

This characteristic plays a crucial role in the convergence analysis. The proof of Lemma 3.1 is presented in Appendix A.7, accompanied by supplementary lemmas in Appendix A.6.

Given the decreasing property of the merit function $V_k$, we can establish the non-asymptotic convergence rate for the sequence $\{(x^k, y^k, \theta^k)\}$ generated by Algorithm 1. Due to the Moreau envelope function constraint, conventional constraint qualifications do not hold at any feasible point of the constrained problem (4), see, e.g., (Ye & Zhu, 1995, Proposition 3.2) and (Gao et al., 2023, Section 2.3). This renders the classical Karush–Kuhn–Tucker (KKT) condition unsuitable as an appropriate necessary optimality condition for problem (4). To circumvent this limitation, inspired by the approximate KKT condition proposed in Helou et al. (2020), which is independent of constraint qualifications, we consider using the residual function as follows,

$$R_k(x, y) := \text{dist}\left(0, \nabla F(x, y) + c_k\left(\nabla f(x, y) + \partial g(x, y) - \nabla v_\gamma(x, y)\right) + \mathcal{N}_{X \times Y}(x, y)\right). \tag{15}$$

This residual function can also be regarded as a stationarity measure for the following penalized version of the constrained problem (4), with $c_k$ serving as the penalty parameter:

$$\min_{(x,y) \in X \times Y} \psi_{c_k}(x, y) := F(x, y) + c_k\left(f(x, y) + g(x, y) - v_\gamma(x, y)\right). \tag{16}$$

It is evident that $R_k(x, y) = 0$ if and only if $0 \in \partial \psi_{c_k}(x, y) + \mathcal{N}_{X \times Y}(x, y)$, i.e., the point $(x, y)$ is a stationary point to the unconstrained problem (16).

**Theorem 3.1.** *Under Assumptions 3.1 and 3.2, suppose $\gamma \in (0, \frac{1}{2\rho_{f_2} + 2\rho_{g_2}})$, $c_k = \underline{c}(k + 1)^p$ with $p \in (0, 1/2)$, $\underline{c} > 0$ and $\eta_k \in [\underline{\eta}, (1/\gamma - \rho_{f_2})/(L_f + 1/\gamma)^2) \cap [\underline{\eta}, 1/\rho_{g_2})$ with $\underline{\eta} > 0$, then there exists $c_\alpha, c_\beta > 0$ such that when $\alpha_k \in (\underline{\alpha}, c_\alpha)$ and $\beta_k \in (\underline{\beta}, c_\beta)$ with $\underline{\alpha}, \underline{\beta} > 0$, the sequence of $(x^k, y^k, \theta^k)$ generated by Algorithm 1 satisfies*

$$\min_{0 \leq k \leq K}\left\|\theta^k - \theta_\gamma^*(x^k, y^k)\right\| = O\left(\frac{1}{K^{1/2}}\right),$$

*and*

$$\min_{0 \leq k \leq K} R_k(x^{k+1}, y^{k+1}) = O\left(\frac{1}{K^{(1-2p)/2}}\right).$$

*Furthermore, if the sequence $\psi_{c_k}(x^k, y^k)$ is upper-bounded, the sequence of $(x^k, y^k)$ satisfies*

$$\varphi(x^K, y^K) - v_\gamma(x^K, y^K) = O\left(\frac{1}{K^p}\right).$$

Table 2: Comparison of total iterative time with representative BLO methods in LL non-convex case with different dimensions.

| Category | Dimension =1 | | | | Dimension =10 | | | |
|---|---|---|---|---|---|---|---|---|
| Methods | BVFIM | BOME | IAPTT | MEHA | BVFIM | BOME | IAPTT | MEHA |
| Time (S) | 101.9 | 3.163 | 32.327 | **0.388** | 33.987 | 7.634 | 233.7 | **5.086** |
| Category | Dimension =50 | | | | Dimension =100 | | | |
| Methods | BVFIM | BOME | IAPTT | MEHA | BVFIM | BOME | IAPTT | MEHA |
| Time (S) | 383.0 | 20.25 | 210.6 | **9.447** | 33.987 | 803.5 | 707.9 | **14.31** |

## 4 EXPERIMENTAL RESULTS

We validate the effectiveness and efficiency of MEHA on various synthetic problems, two typical hyper-parameter learning tasks and the real-world neural architecture search application. We compare MEHA against Explicit Gradient Based Methods (EGBMs, including RHG Franceschi et al. (2017), BDA Liu et al. (2020) and IAPTT Liu et al. (2021c)), Implicit Gradient Based Methods (IGBMs, CG Pedregosa (2016) and NS Rajeswaran et al. (2019)) and currently methods (BV-FIM Liu et al. (2021b), BOME Ye et al. (2022), F²SA Kwon et al. (2023), BAMM Liu et al. (2023b)). These experiments collectively substantiate the superior performance of MEHA.

### 4.1 SYNTHETIC NUMERICAL VERIFICATION

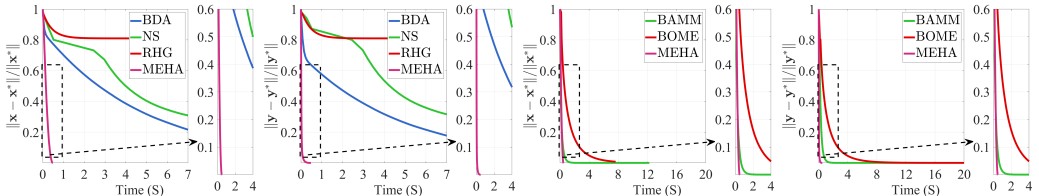

Figure 1: Illustrating the convergence curves of advanced BLO schemes and MEHA by the criterion of $\|x - x^*\|/\|x^*\|$ and $\|y - y^*\|/\|y^*\|$ in LL merely convex case.

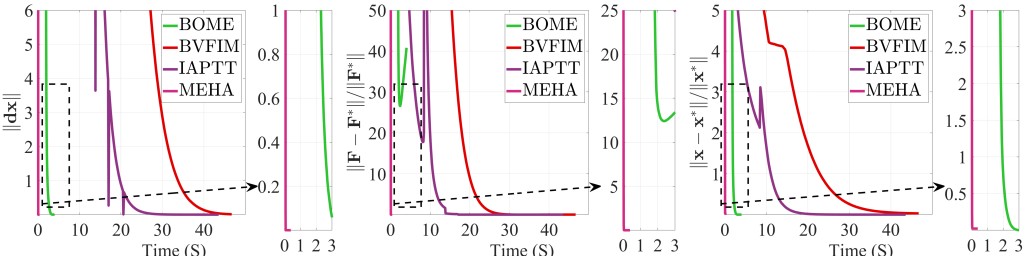

Figure 2: Visualizing the convergence behavior of BOME, BVFIM, IAPTT, and MEHA in LL non-convex case with one dimension. We use metrics such as the descent direction $\|\mathbf{d}x\|$, UL objective $F$, and the reconstruction error with $x$ for comparison.

**LL Merely Convex Case.** We demonstrate the high efficiency of MEHA in LL merely convex case using the following toy example:

$$\min_{x,y_1,y_2 \in \mathbb{R}^n} \frac{1}{2}\|x - y_2\|^2 + \frac{1}{2}\|y_1 - e\|^2 \text{ s.t. } y = (y_1, y_2) \in \operatorname{argmin}_{y \in \mathbb{R}^{2n}} \frac{1}{2}\|y_1\|^2 - x^\top y_1. \quad (17)$$

In concrete, the unique solution of the above objective is $(e, e, e) \in \mathbb{R}^{3n}$. We plot the convergence behavior in Figure 1. Specifically, MEHA reduces the number of iterative steps by 66% and inference time by 87.4% compared to the latest BAMM scheme.

Table 3: Comparison of LL non-smooth case utilizing lasso regression under diverse dimensions.

| | Dimension=2 | | | | Dimension=100 | | | Dimension=1000 | | |
|---|---|---|---|---|---|---|---|---|---|---|
| | Grid | Random | TPE | MEHA | Random | TPE | MEHA | Random | TPE | MEHA |
| Time (S) | 14.87 | 17.11 | 3.32 | **1.91** | 86.27 | 232.61 | **2.67** | 700.74 | 2244.44 | **26.22** |
| Promotion | ×37.0 | ×8.96 | ×1.74 | - | ×32.3 | ×87.1 | - | ×26.7 | ×85.6 | - |

**LL Non-Convex Case.** To demonstrate the wide applicability and superiority of MEHA, we also showcase its effectiveness in LL non-convex case, using the following example:

$$\min_{x\in\mathbb{R}, y\in\mathbb{R}^n} \|x - a\|^2 + \|y - a\boldsymbol{e} - \boldsymbol{c}\|^2 \text{ s.t. } y_i \in \arg\min_{y_i\in\mathbb{R}} \sin(x + y_i - c_i) \ \forall i, \quad (18)$$

where $\boldsymbol{c} \in \mathbb{R}^n$ and $a \in \mathbb{R}$. Following the literature Liu et al. (2021b), we can obtain the optimal solution as $x^* = \frac{(1-n)a+nC}{1+n}$ and $y_i^* = C + c_i - x^* \ \forall i$. Here $C = \arg\min_{C_k} \{\|C_k - 2a\| : C_k = -\frac{\pi}{2} + 2k\pi, k \in \mathbb{Z}\}$. The optimal UL objective is $F^* = \frac{n(C-2a)^2}{1+n}$. When $n = 1$, given $a = 2$ and $\boldsymbol{c} = 2$, the concrete solution is $x^* = 3\pi/4$ and $y^* = 3\pi/4 + 2$. Given the initialization point $(x_0, y_0) = (-6, 0)$, we compare the performance with a few capable BLO schemes (e.g., BVFIM, BOME, and IAPTT) with different dimensions. In terms of numerical comparison, we provide the iterative times along with advanced competitors in Table 2. In Figure 2, we present convergence curves in the non-convex scenario, using different metrics. Additionally, in Figure 3, we examine their performance across various dimensions. Firstly, MEHA and BOME exhibit the fastest convergence speed, leveraging Hessian-free computation. Secondly, MEHA outperforms BOME in approximating the optimal UL objective, as demonstrated in the second subFigure Furthermore, we highlight the computational efficiency of MEHA on a large scale in Figure 6 in Appendix A.2.

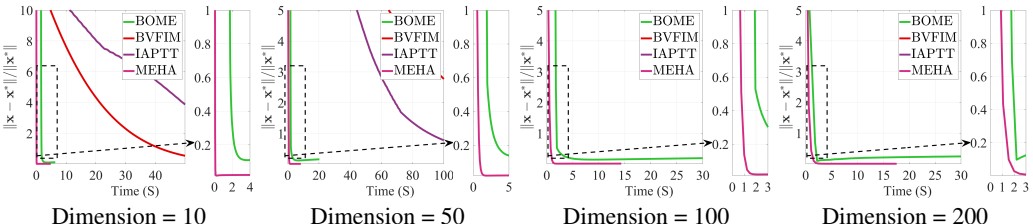

Dimension = 10          Dimension = 50          Dimension = 100          Dimension = 200

Figure 3: Illustrating the convergence curves of advanced BLO and MEHA by the criterion of $\|x - x^*\|/\|x^*\|$ in LL non-convex case with different dimensions.

**LL Non-Smooth Case.** We employ conventional lasso regression on synthetic data to verify the superiority of MEHA, formulated as follows:

$$\min_{x\in\mathbb{R}^n, 0\leq x\leq 1, y\in\mathbb{R}^n} \sum_{i=1}^n y_i \quad \text{s.t.} \quad y \in \arg\min_{y\in\mathbb{R}^n} \frac{1}{2}\|y - \boldsymbol{a}\|^2 + \sum_{i=1}^n x_i\|y_i\|_1, \quad (19)$$

where $\boldsymbol{a} := \left(\frac{1}{n}, \frac{1}{n}, \cdots, \frac{1}{n}, -\frac{1}{n}, -\frac{1}{n}, \cdots -\frac{1}{n}\right) \in \mathbb{R}^n$. The number of positive and negative values is $\frac{n}{2}$. The optimal solution can be calculated as $x_i \in [\frac{1}{n}, 1]$, $y_i = 0$ when $i = 1, \cdots \frac{n}{2}$ and $x_i = 0$, $y_i = -\frac{1}{n}$ when $i = \frac{n}{2} + 1, \cdots n$. The solving time of different methods with various dimensions is reported in Table 3. Compared to grid search, random search, and Bayesian optimization-based TPE Bergstra et al. (2013), MEHA consistently requires the least amount of time to find the optimal solutions across various dimensions, particularly in large-scale scenarios (e.g., 1000 dimension). You can examine the convergence curves of MEHA in Figure 7 in Appendix A.2.

### 4.2 REAL-WORLD APPLICATIONS

**Few-Shot Learning.** The objective of N-way M-shot classification is to enhance the adaptability of the hyper model, enabling it to quickly adapt to new tasks. In our experimental analysis, we conducted experiments on the Omniglot dataset Finn et al. (2017), specifically in 10-way 1-shot and 20-way 1-shot scenarios. Table 4 presents a comparison of the run-time required to achieve the same accuracy levels (90% for both 10-way and 20-way scenarios). Notably, MEHA achieves similar accuracy levels while significantly reducing computational time.

Table 4: Comparison of the results for few-shot learning (90% for 10-way and 20-way) and data hyper-cleaning tasks (FashionMINIST and MNIST datasets).

| Method | 10-Way | | 20-Way | | FashionMNIST | | | MNIST | | |
|---|---|---|---|---|---|---|---|---|---|---|
| | Acc. (%) | Time (S) | Acc. (%) | Time (S) | Acc. (%) | Time (S) | Steps | Acc. (%) | Time (S) | Steps |
| RHG | 89.77 | 557.71 | 90.17 | 550.51 | 81.07 | 50.245 | 460 | 87.04 | 30.743 | 270 |
| BDA | 89.61 | 869.98 | 89.78 | 1187.63 | 81.06 | 55.656 | 140 | 87.06 | 75.321 | 380 |
| CG | 89.56 | 363.76 | 89.39 | 602.24 | 81.00 | 9.705 | 150 | 87.06 | 26.947 | 410 |
| BAMM | 90.57 | 180.48 | 90.13 | 255.99 | 81.02 | 6.193 | 630 | 87.02 | 3.797 | 380 |
| MEHA | 90.45 | **150.39** | 90.87 | **246.72** | 81.07 | **3.297** | 630 | 87.03 | **3.063** | 530 |

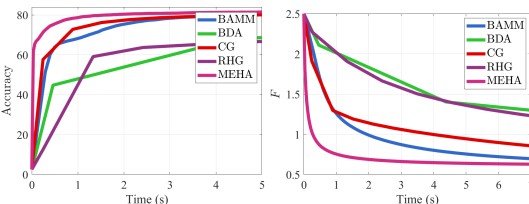

Figure 4: Comparison of the validation loss $F$ and accuracy for hyper-cleaning on FashionMINIST dataset.

**Data Hyper-Cleaning.** In the right part of Table 4, we present the accuracy and time required for various methods to achieve similar levels of accuracy (81% for FashionMNIST and 87% for MNIST) in data hyper-cleaning tasks. Notably, MEHA substantially reduces the time needed to reach the desired solution. Additionally, in Figure 4, we visualize the validation loss and test accuracy for different methods. It is evident that MEHA exhibits the fastest convergence rate and maintains its superior speed even as it rapidly achieves an accuracy rate of 81%.

**Neural Architecture Search.** The primary objective is to discover high-performance neural network structures through an automated process. We specifically focus on gradient-based differentiable NAS methods (e.g., DARTS Liu et al. (2018)), which represent a typical LL non-convex case. To illustrate the consistency of the discovered architecture's performance, we provide accuracy results at various stages in Table 5. It becomes evident that MEHA consistently exhibits superior performance at different stages compared to specialized designs for architecture search. It is worth noting that these advanced NAS methods utilize specialized techniques (e.g., progressive search and channel splitting) to accelerate the searching procedure. By leveraging these effective techniques, we can further reduce the search cost, such as time. Furthermore, we provide a comparison with existing BLO methods, demonstrating our superiority in addressing real-world large-scale LL non-convex applications.

Table 5: Comparing Top-1 accuracy in searching, inference, and final test stages for DARTS Liu et al. (2018), P-DARTS Chen et al. (2019), PC-DARTS Xu et al. (2019), and BLO schemes.

| Methods | Searching | | Inference | | Test | Params (M) |
|---|---|---|---|---|---|---|
| | Train | Valid | Train | Valid | | |
| DARTS | 98.320 | 88.940 | 99.481 | 95.639 | 95.569 | 1.277 |
| P-DARTS | 96.168 | 90.488 | 99.802 | 95.701 | 95.710 | 1.359 |
| PC-DARTS | 84.821 | 83.516 | 98.163 | 95.630 | 95.540 | 1.570 |
| RHG | 98.448 | 89.556 | 99.688 | 95.340 | 95.340 | 1.359 |
| CG | 99.126 | 89.298 | 98.909 | 95.499 | 95.370 | 1.268 |
| IAPTT | 98.904 | 99.512 | 99.776 | 95.840 | 95.809 | 1.963 |
| MEHA | 98.936 | **99.716** | 99.601 | **96.101** | **95.860** | 1.406 |

## 5    CONCLUSIONS

By rethinking the Moreau envelope-based reformulation for general nonconvex BLOs, we propose a provably single-loop and Hessian-free gradient-based algorithm, named MEHA. We validate the effectiveness and efficiency of MEHA for large-scale nonconvex-nonconvex BLO in deep learning. By leveraging the simplicity of our approach and integrating techniques such as variance reduction and momentum, we would be interested in studying stochastic algorithms in the future.

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

## A    APPENDIX

The appendix is organized as follows:

- Expanded related work is provided in Section A.1.
- Additional experimental results are provided in Section A.2.
- Experimental details are provided in Section A.3.
- The equivalent result of the reformulated problem 4 is provided in Section A.4.
- We prove the weakly convexity and derive the gradient formula of Moreau Envelope in Section A.5.
- Some useful auxiliary lemmas are provided in Section A.6.
- The proof of Lemma 3.1 is given in Section A.7.
- The proof of Proposition 3.1 is provided in Section A.8.

### A.1    EXPANDED RELATED WORK

In this section, we provide an extensive review of recent studies closely related to ours.

**Nonconvex-Convex BLO.** The LL strong convexity significantly contributes to the development of efficient BLO algorithms, see, e.g., Ghadimi & Wang (2018); Hong et al. (2023); Chen et al. (2021); Ji et al. (2021); Chen et al. (2022a); Ji et al. (2021; 2022); Kwon et al. (2023). It guarantees the uniqueness of the LL minimizer (Lower-Level Singleton), which facilitates the demonstration of asymptotic convergence for the iterative differentiation-based approach Franceschi et al. (2018). If further the LL objective is twice differentiable, the gradient of the UL objective (hyper-gradient) can be expressed using the implicit function theorem. Then the uniformly LL strong convexity implies both the smoothness and the Lipschitz continuity properties of the LL solution mapping. These essential properties facilitates the demonstration of non-asymptotic convergence for both the iterative differentiation and the approximate implicit differentiation approaches with rapid convergence rates, see e.g., Ghadimi & Wang (2018); Hong et al. (2023); Chen et al. (2021); Yang et al. (2021); Ji & Liang (2022); Ji et al. (2021); Khanduri et al. (2021); Sow et al. (2022b); Ji et al. (2022); Sow et al. (2022b); Arbel & Mairal (2022a); Li et al. (2022); Dagréou et al. (2022); Yang et al. (2023). Due to the implicit gradient, the methods mentioned above necessitate costly manipulation involving the Hessian matrix, making them all second-order methods. Recently, Kwon et al. (2023) developed stochastic and deterministic fully first-order BLO algorithms based on the value function approach Ye & Zhu (1995), and established their non-asymptotic convergence guarantees, while an improved convergence analysis is provided in the recent work Chen et al. (2023).

In the absence of strong convexity, additional challenges may arise, including the presence of multiple LL solutions (Non-Singleton), which can hinder the application of implicit-based approaches involved in the study of nonconvex- strongly-convex BLOs. To tackle Non-Singleton, sequential averaging methods (also referred to as aggregation methods) were proposed in Liu et al. (2020); Li et al. (2020); Liu et al. (2022; 2023b). Recent advances include value function based difference-of-convex algorithm Ye et al. (2023); Gao et al. (2022); primal-dual algorithms Sow et al. (2022a); first-order penalty methods using a novel minimax optimization reformulation Lu & Mei (2023).

**Nonconvex-Nonconvex BLO.** While the nonconvex-convex BLO has been extensively studied in the literature, the efficient methods for nonconvex-nonconvex BLO remain under-explored. Beyond the LL convexity, the authors in Liu et al. (2021c) develop a method with initialization auxiliary and pessimistic trajectory truncation; the study Arbel & Mairal (2022b) extends implicit differentiation to a class of nonconvex LL functions with possibly degenerate critical points and then develops unrolled optimization algorithms. However, these works requires second-order gradient information and do not provide finite-time convergence guarantees. Still with the second-order gradient information but providing non-asymptotic analysis, the recent works Huang (2023) and Xiao et al. (2023) propose a momentum-based BLO algorithm and a generalized alternating method for BLO with a nonconvex LL objective that satisfies the Polyak-Łojasiewicz (PL) condition, respectively. In contrast to these methods discussed above, the value function reformulation of BLO was firstly utilized in Liu et al. (2021b) to develop BLO algorithms in machine learning, using an interior-point method

combined with a smoothed approximation. But it lacks a complete non-asymptotic analysis. Subsequently, Ye et al. (2022) introduced a fully first-order value function based BLO algorithm. They also established the non-asymptotic convergence results when the LL objective satisfies the PL or local PL conditions. Recently, Shen & Chen (2023) proposed a penalty-based fully first-order BLO algorithm and established its finite-time convergence under the PL conditions. Notably, this work relaxed the relatively restrictive assumption on the boundedness of both the UL and LL objectives that was present in Ye et al. (2022).

**Nonsmooth BLO.** Despite plenty of research focusing on smooth BLOs, there are relatively fewer studies addressing nonsmooth BLOs, see, e.g., Mairal et al. (2011); Okuno et al. (2018); Bertrand et al. (2020; 2022). However, these works typically deal with special nonsmooth LL problems, e.g., task-driven dictionary learning with elastic-net (involving the $\ell_1$-norm) in Mairal et al. (2011); the Lasso-type models (including the $\ell_1$-norm as well) for hyper-parameter optimization in Bertrand et al. (2020); $\ell_p$-hyperparameter learning with $0 < p < 1$ in Okuno et al. (2018); non-smooth convex learning with separable non-smooth terms in Bertrand et al. (2022). Recently, there is a number of works studying BLOs with general nonsmooth LL problems. By decoupling hyperparameters from the regularization, based on the value function approach, Gao et al. (2022) develop a sequentially convergent Value Function-based Difference-of-Convex Algorithm with inexactness for a specific class of bi-level hyper-parameter selection problems. Gao et al. (2023) introduces a Moreau envelope-based reformulation of BLOs and develops an inexact proximal Difference-of-weakly-Convex algorithm with sequential convergence, to substantially weaken the underlying assumption in Ye et al. (2023) from lower level full convexity to weak convexity. There is also a line of works devoted to tackle the nonsmooth UL setting, including: Bregman distance-based method in Huang et al. (2022); proximal gradient-type algorithm in Chen et al. (2022b).

## A.2  Additional Experimental Results

**LL Strong Convex Case.** We first illustrate the convergence results leveraging the toy numerical problem in BDA Liu et al. (2020) with a lower-level convex objective, which can be written as:

$$\min_{x \in \mathbb{R}^n} \frac{1}{2} \|x - z_0\|^2 + \frac{1}{2} y^*(x)^\top A y^*(x) \text{ s.t. } y^*(x) = \arg \min_{y \in \mathbb{R}^n} f(x, y) = \frac{1}{2} y^\top A y - x^\top y, \quad (20)$$

where $x \in \mathbb{R}^n$ and $y \in \mathbb{R}^n$. We define $A$ has the positive-definite symmetric property and $A \in \mathbb{S}^{n \times n}$, $z_0 \neq 0$ and $z_0 \in \mathbb{R}^n$. Concretely, we set $A = I$ and $z_0 = e$. Thus, the optimal solution is $x^* = y^* = e/2$, where $e$ represents the vector containing all elements equal to one. Obviously, this case satisfies the most convergence assumptions including Explicit Gradient Based Methods (EGBMs, including RHG Franceschi et al. (2017), BDA and IAPTT Liu et al. (2021c)), Implicit Gradient Based Methods (IGBMs, CG Pedregosa (2016) and NS Rajeswaran et al. (2019)) and current proposed methods (BRC Liu et al. (2023a), BOME Ye et al. (2022), F2SA Kwon et al. (2023), BAMM Liu et al. (2023b)). In details, the numerical comparisons are reported in Table 6. We also compare these methods in Figure 5. From the behaviors in Figure 5, our method achieves the fastest convergences compared with EGBMs, IGBMs, and single-loop methods in Table 6, it can be clearly observed that MEHA has two significant promotions. Compared with the effective BAMM, MEHA achieves 85.8% improvement of inference time. Furthermore, the proposed scheme has the lowest computational cost, which only exploits 10.35% memory of BAMM.

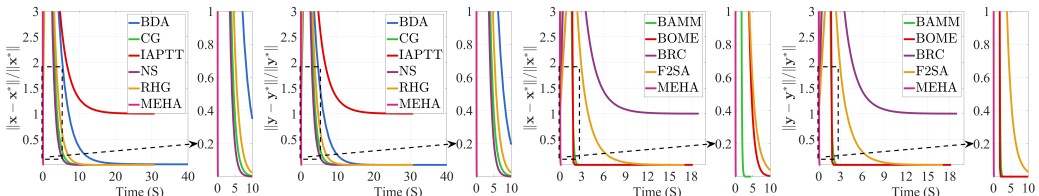

Figure 5: Illustrating the convergence curves of advanced BLO methods and MEHA by the criterion of $\|x - x^*\|/\|x^*\|$ and $\|y - y^*\|/\|y^*\|$ in LL strong convex case.

Table 6: Basic properties of time and memory in LL strong convex case.

| Category | EGBMs | | | IGBMs | | Others | | | | |
|---|---|---|---|---|---|---|---|---|---|---|
| Methods | RHG | BDA | IAPTT | CG | NS | BRC | BOME | F$^2$SA | BAMM | MEHA |
| Time (S) | 14.01 | 50.13 | 31.98 | 15.66 | 21.10 | 19.48 | 11.18 | 16.21 | 2.650 | **0.375** |
| Memory | 160768 | 212480 | 160768 | 111104 | 110592 | 12800 | 14848 | 12288 | 14848 | **1536** |

Table 7: Computational efficiency comparison of BLO schemes in LL strong-convex case.

| Convergence Time (Dimension=10000) | | | | | | Convergence Time (Dimension=30000) | | | | | |
|---|---|---|---|---|---|---|---|---|---|---|---|
| | RHG | CG | NS | BAMM | MEHA | | RHG | CG | NS | BAMM | MEHA |
| Time (S) | 70.59 | 36.97 | 40.53 | 5.890 | 2.352 | Time (S) | 110.26 | 64.21 | 65.73 | 5.949 | 3.538 |
| Promotion | ×30.0 | ×15.7 | ×17.2 | ×2.5 | - | Promotion | ×31.2 | ×18.2 | ×18.6 | ×1.68 | - |

**Computational Efficiency Under Large Scale.** We provide the convergence time compared with advanced BLO method in LL non-convex case, as shown in Figure 6. We can obviously observe that, our method realize the fast convergence among different dimensions, especially for large-scale computation. We evaluate the computational efficiency to by increasing the dimension of $x$ and $y$

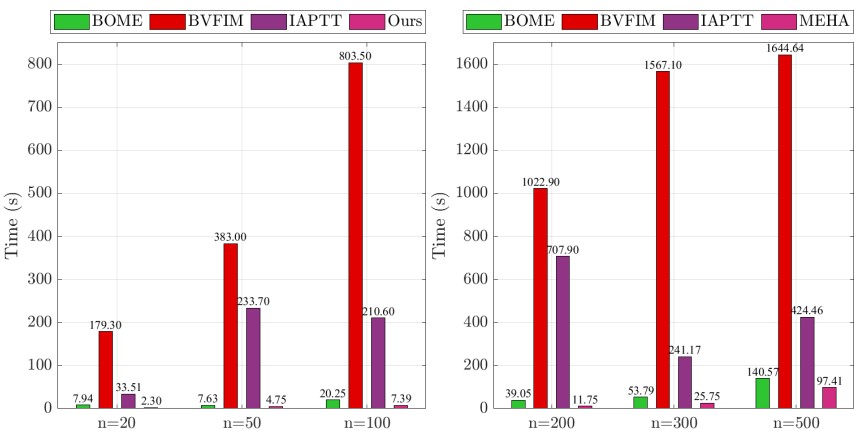

Figure 6: Computational efficiency comparison of advanced BLO schemes in LL non-convex case.

as $10^4$ and $3 \times 10^4$ in LL strong convex case. Table 7 illustrates the superiority of our method to handle BLO problems with large dimension.

**LL Non-Smooth Case.** We provide the convergence curves of our method MEHA on different dimensions in Figure 7. We can conclude that our method can find the optimal hyper-parameter $x^*$ effectively and the optimal solution $\{x^*, y^*\}$ under diverse high dimensions.

### A.3 EXPERIMENTAL DETAILS

We conducted the experiments on a PC with Intel i7-8700 CPU (3.2 GHz), 32GB RAM and NVIDIA RTX 1070 GPU. We utilized the PyTorch framework on the 64-bit Windows system.

**Synthetic Numerical Examples.** The hyper-parameter settings for diverse numerical experiments are summarized in Table 8. Specifically, as for the compared methods in LL non-smooth case, we follow the effective practice Gao et al. (2022). We utilize the SGD optimizer to update UL variable $x$. We uniformly utilize the $\|x - x^*\|/\|x^*\| \leq 10^{-2}$ criterion in LL merely convex case and loss $\|F^k - F^{k-1}\|/\|F^k\| \leq 2 \times 10^{-4}$ in LL non-convex case. The learning steps $\alpha$, $\beta$ and $\eta$ are fixed. $p$ is setted as 0.49 for the update of $c_k$.

**Few-shot Learning.** As for this task, the upper-level variables $x$ represent the shared weights for feature extraction. $y$ denotes the task-specific parameters. Leveraging the cross-entropy loss as the

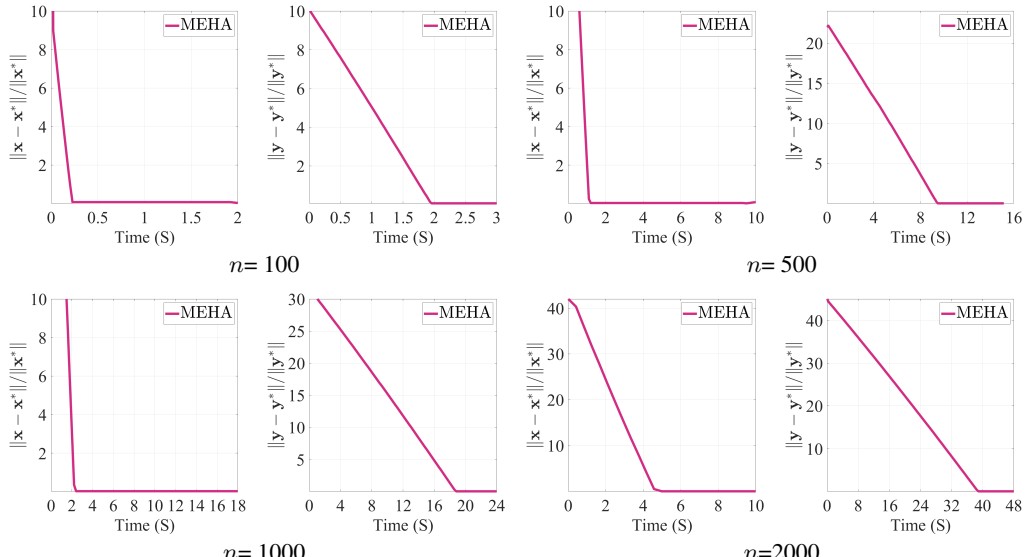

Figure 7: Illustrating the convergence curves by the criterion of $\|x - x^*\|/\|x^*\|$ and $\|y - y^*\|/\|y^*\|$ in LL non-smooth case with different dimensions.

Table 8: Values for hyper-parameters of synthetic numerical experiments.

| Category | $\alpha_0$ | $\beta_0$ | $\eta_0$ | $\gamma$ | $\underline{c}$ |
|---|---|---|---|---|---|
| LL strong-convex | 1.5 | 0.8 | 0.8 | 10 | 33.3 |
| LL merely-convex | 0.012 | 0.1 | 0.009 | 5 | 0.167 |
| LL non-convex | 5e−4 | 5e−4 | 0.001 | 200 | 0.02 |
| LL non-smooth | 0.1 | 1e−5 | 0.1 | 10 | 2 |

objective $\mathcal{L}$, we provide the bi-level formulation as:

$$\min_x \sum_j \mathcal{L}(x, y^j; \mathcal{D}_{\texttt{val}}^j) \quad \text{s.t.} \quad y = \arg\min_y \mathcal{L}(x, y^j; \mathcal{D}_{\texttt{train}}^j). \tag{21}$$

Following with the practice Liu et al. (2023c), we utilize four layers of convolution blocks (ConvNet-4) to construct the backbone (i.e., $x$), which is widely utilized for few-shot learning tasks. The task-specific classifier $y$ is composited by fully-connection layers with softmax operation. Adam and SGD optimizers are utilized to update $x$ and $y$ for all algorithms fairly. The concrete hyper-parameters of Alg.1 and other shared hyper-parameters are summarized in Table 9. We utilize the inverse power learning rate annealing strategy to dynamically adjust the learning rate ($\alpha$ and $\beta$). $\eta$ and $\gamma$ are fixed.

Table 9: Values for hyper-parameters of few-shot learning.

| Parameter | Meta batch size | Hidden size | $\alpha_0$ | $\beta_0$ | $\eta_0$ | $\gamma$ | $\underline{c}$ |
|---|---|---|---|---|---|---|---|
| Value | 16 | 32 | 0.008 | 0.1 | 0.001 | 100 | 0.5 |

**Data Hyper-Cleaning.** The mathematical formulation can be written as:

$$\min_x \sum_{\mathbf{u}_i, \mathbf{v}_i \in \mathcal{D}_{\texttt{val}}} \mathcal{L}(y(x); \mathbf{u}_i, \mathbf{v}_i) \quad \text{s.t.} \quad y = \arg\min_y \sum_{\mathbf{u}_i, \mathbf{v}_i \in \mathcal{D}_{\texttt{train}}} [\sigma(x)]_i \mathcal{L}(y; \mathbf{u}_i, \mathbf{v}_i), \tag{22}$$

where the upper-level variable $x$ is a vector with the same dimension of the number of corrupted examples. $y$ denotes the target classification model. $\sigma(x)$ is a sigmoid function. $\{\mathbf{u}, \mathbf{v}\}$ are the data pairs. In detail, we only utilize one-layer of fully-connection to define $y$. Two datasets FashionMNIST and MNIST are utilized to conduct the experiments. We randomly split these datasets to composite the training, validation and testing subsets with 5000, 5000, 10000 examples, respectively. Half of data in the training dataset is tampered. The concrete hyper-parameters of Alg.1 are

summarized in Table 10. Adam optimizer is utilized to update the UL variable $x$ fairly. We utilize the inverse power learning rate annealing strategy to dynamically adjust the learning rate ($\alpha$ and $\beta$). $\eta$ and $\gamma$ are fixed.

Table 10: Values for hyper-parameters of data hyper-cleaning.

| Parameter | $\alpha_0$ | $\beta_0$ | $\eta_0$ | $\gamma$ | $\underline{c}$ |
|---|---|---|---|---|---|
| Value | 0.01 | 0.1 | 0.0004 | 10 | 40 |

**Neural Architecture Search.** The bi-level formulation of neural architecture search is

$$\min_x \mathcal{L}_{\text{val}}(y^*(x), x; \mathcal{D}_{\text{val}}) \quad \text{s.t.} \quad y^*(x) = \arg\min_y \mathcal{L}_{\text{train}}(y, x; \mathcal{D}_{\text{train}}), \tag{23}$$

where the architecture parameters are denoted as the upper-level variable $x$ and the lower-level variable $y$ represents the network weights. $\mathcal{L}_{\text{val}}$ and $\mathcal{L}_{\text{train}}$ are the losses on validation and training datasets. The definition of search space, cells, and experimental hyper-parameters settings are following with the literature Liu et al. (2018). We leveraged the Cifar-10 dataset to perform the experiments of image classification. As for the super-network, we conducted the search procedure with three layers of cells for 50 epochs. The network for training is increased with 8 layers and trained from scratch with 600 epochs. The concrete hyper-parameters of Alg.1 are summarized in Table 11. We utilized the cosine decreasing learning rate annealing strategy to dynamically adjust the learning rate ($\alpha$, $\beta$ and $\eta$).

Table 11: Values for hyper-parameters of neural architecture search.

| Parameter | $\alpha_0$ | $\beta_0$ | $\eta_0$ | $\gamma$ | $\underline{c}$ |
|---|---|---|---|---|---|
| Value | 8e-5 | 0.025 | 0.025 | 200 | 2 |

## A.4 EQUIVALENCE OF MOREAU ENVELOPE BASED REFORMULATION

The following theorem establishes the equivalence between the Moreau Envelope based reformulation problem (4) and the relaxed bilevel optimization problem (5). The proof is inspired by the one of Theorem 2.1 in Gao et al. (2023). For the convenience of the reader, we restate problems (4) and (5) as follows:

$$\min_{(x,y)\in X\times Y} F(x,y) \quad \text{s.t.} \quad \varphi(x,y) - v_\gamma(x,y) \le 0, \tag{4}$$

where $v_\gamma(x,y) := \inf_{\theta\in Y}\left\{\varphi(x,\theta) + \frac{1}{2\gamma}\|\theta - y\|^2\right\}$, $\varphi(x,y) = f(x,y) + g(x,y)$, and

$$\min_{x\in X, y\in Y} F(x,y) \quad \text{s.t.} \quad 0 \in \nabla_y f(x,y) + \partial_y g(x,y) + \mathcal{N}_Y(y). \tag{5}$$

**Theorem A.1.** *Suppose that $\varphi(x,\cdot)$ is $\rho_{\varphi_2}$-weakly convex on $Y$ for all $x$, i.e., $\varphi(x,\cdot) + \frac{\rho_{\varphi_2}}{2}\|\cdot\|^2$ is convex on $Y$ for all $x$. Then for $\gamma \in (0, 1/\rho_{\varphi_2})$, the Moreau Envelope based reformulation problem (4) is equivalent to the relaxed BLO problem (5).*

*Proof.* First, given any feasible point $(x,y)$ of problem (4), it necessarily belongs to $X \times Y$ and satisfies

$$\varphi(x,y) \le v_\gamma(x,y) := \inf_{\theta\in Y}\left\{\varphi(x,\theta) + \frac{1}{2\gamma}\|\theta - y\|^2\right\} \le \varphi(x,y).$$

From which, it follows that $\varphi(x,y) = v_\gamma(x,y)$ and thus $y \in \operatorname{argmin}_{\theta\in Y}\left\{\varphi(x,\theta) + \frac{1}{2\gamma}\|\theta - y\|^2\right\}$. This leads to

$$0 \in \nabla_y f(x,y) + \partial_y g(x,y) + \mathcal{N}_Y(y),$$

implying that $(x,y)$ is feasible for problem (5).

Conversely, consider that $(x,y)$ is an feasible point of problem (5). This implies that $(x,y) \in X \times Y$, $0 \in \nabla_y f(x,y) + \partial_y g(x,y) + \mathcal{N}_Y(y)$. Given that $\varphi(x,\cdot) : \mathbb{R}^m \to \mathbb{R}$ is $\rho_{\varphi_2}$-weakly convex on $Y$, then when $\gamma \in (0, 1/\rho_{\varphi_2})$, the function $\varphi(x,\cdot) + \frac{1}{2\gamma}\|\cdot - y\|^2$ is convex on $Y$, making it lower regular.

Clearly, $\delta_Y(\cdot)$ is lower regular since $Y$ is a closed convex set. By leveraging the subdifferential sum rules for two lower regular l.s.c. functions (Mordukhovich, 2018, Theorem 2.19), we arrive at

$$\partial\left(\varphi(x,\cdot) + \frac{1}{2\gamma}\|\cdot - y\|^2 + \delta_Y(\cdot)\right) = \nabla_y f(x,\cdot) + \partial_y g(x,\cdot) + (\cdot - y)/\gamma + \mathcal{N}_Y(\cdot).$$

With the right-hand set-valued mapping at $y$ containing 0, we can deduce that

$$0 \in \partial_\theta\left(\varphi(x,\theta) + \frac{1}{2\gamma}\|\theta - y\|^2 + \delta_Y(\theta)\right)\bigg|_{\theta=y}.$$

Thus, invoking the first-order optimally condition for convex functions, we infer

$$y \in \operatorname{argmin}_{\theta \in Y}\left\{\varphi(x,\theta) + \frac{1}{2\gamma}\|\theta - y\|^2\right\}.$$

This implies $\varphi(x,y) = v_\gamma(x,y)$, confirming $(x,y)$ as an feasible point to problem (4). $\qquad\square$

## A.5 Properties of Moreau Envelope

By invoking (Rockafellar, 1974, Theorem 1), we have that when the LL problem is fully convex, the Moreau Envelope $v_\gamma(x,y)$ is also convex. We further generalize this finding in the subsequent lemma, showing that $v_\gamma(x,y)$ retains weak convexity when the LL problem exhibits weak convexity. The foundation for this proof draws inspiration from Theorem 2.2 as presented in Gao et al. (2023).

**Lemma A.1.** *Suppose that $\varphi(x,y)$ is $(\rho_{\varphi_1}, \rho_{\varphi_2})$-weakly convex on $X \times Y$. Then for $\gamma \in (0, \frac{1}{2\rho_{\varphi_2}})$, $\rho_{v_1} \geq \rho_{\varphi_1}$ and $\rho_{v_2} \geq \frac{1}{\gamma}$, the function*

$$v_\gamma(x,y) + \frac{\rho_{v_1}}{2}\|x\|^2 + \frac{\rho_{v_2}}{2}\|y\|^2$$

*is convex on $X \times \mathbb{R}^m$.*

*Proof.* We first extend the definition of the Moreau envelope $v_\gamma(x,y)$ from $x \in X$ to $x \in \mathbb{R}^n$ by

$$v_\gamma(x,y) := \inf_{\theta \in \mathbb{R}^m}\left\{\varphi(x,\theta) + \frac{1}{2\gamma}\|\theta - y\|^2 + \delta_{X \times Y}(x,\theta)\right\} \quad \forall x \in \mathbb{R}^n, y \in \mathbb{R}^m.$$

It follows that $v_\gamma(x,y) = +\infty$ for $x \notin X$. For any $\rho_{v_1}, \rho_{v_2} > 0$, the function $v_\gamma(x,y) + \frac{\rho_{v_1}}{2}\|x\|^2 + \frac{\rho_{v_2}}{2}\|y\|^2$ can be rewritten as

$$\begin{aligned}
&v_\gamma(x,y) + \frac{\rho_{v_1}}{2}\|x\|^2 + \frac{\rho_{v_2}}{2}\|y\|^2 \\
&= \inf_{\theta \in \mathbb{R}^m}\left\{\phi_{\gamma,\rho_v}(x,y,\theta) := \varphi(x,\theta) + \frac{\rho_{v_1}}{2}\|x\|^2 + \frac{\rho_{v_2}}{2}\|y\|^2 + \frac{1}{2\gamma}\|\theta - y\|^2 + \delta_{X \times Y}(x,\theta)\right\}.
\end{aligned}$$

By direct computations, we obtain the following equation,

$$\begin{aligned}
&\phi_{\gamma,\rho_v}(x,y,\theta) \\
&= \varphi(x,\theta) + \frac{\rho_{v_1}}{2}\|x\|^2 + \frac{\rho_{\varphi_2}}{2}\|\theta\|^2 + \delta_{X \times Y}(x,\theta) + \left(\frac{1}{2\gamma} - \frac{\rho_{\varphi_2}}{2}\right)\|\theta\|^2 + \frac{1 + \gamma\rho_{v_2}}{2\gamma}\|y\|^2 - \frac{1}{\gamma}\langle\theta, y\rangle.
\end{aligned}$$

Given that $\rho_{v_1} \geq \rho_{\varphi_1}$, the convexity of $\varphi(x,\theta) + \frac{\rho_{v_1}}{2}\|x\|^2 + \frac{\rho_{\varphi_2}}{2}\|\theta\|^2 + \delta_{X \times Y}(x,\theta)$ can be immediately inferred, given that $\varphi(x,y)$ is $(\rho_{\varphi_1}, \rho_{\varphi_2})$-weakly convex on $X \times Y$.

Further, when $\gamma \in (0, \frac{1}{2\rho_{\varphi_2}})$ and $\rho_{v_2} \geq \frac{1}{\gamma}$, it can be shown that both conditions, $\frac{1}{4\gamma} - \frac{\rho_{\varphi_2}}{2} > 0$ and $\frac{1 + \gamma\rho_{v_2}}{2} \geq 1$, hold. This implies that the function

$$\begin{aligned}
&\left(\frac{1}{2\gamma} - \frac{\rho_{\varphi_2}}{2}\right)\|\theta\|^2 + \frac{1 + \gamma\rho_{v_2}}{2\gamma}\|y\|^2 - \frac{1}{\gamma}\langle\theta, y\rangle \\
&= \left(\frac{1}{4\gamma} - \frac{\rho_{\varphi_2}}{2}\right)\|\theta\|^2 + \frac{1}{\gamma}\left(\frac{1}{4}\|\theta\|^2 + \frac{1 + \gamma\rho_{v_2}}{2}\|y\|^2 - \langle\theta, y\rangle\right),
\end{aligned}$$

is convex with respect to $(y, \theta)$. Therefore, under the conditions $\gamma \in (0, \frac{1}{2\rho_{\varphi_2}})$, $\rho_{v_1} \geq \rho_{\varphi_1}$ and $\rho_{v_2} \geq \frac{1}{\gamma}$, the extended-valued function $\phi_{\gamma,\rho_v}(x, y, \theta)$ is convex with respect to $(x, y, \theta)$ over $\mathbb{R}^n \times \mathbb{R}^m \times \mathbb{R}^m$. This, in turn , establishes the convexity of

$$v_\gamma(x, y) + \frac{\rho_{v_1}}{2}\|x\|^2 + \frac{\rho_{v_2}}{2}\|y\|^2 = \inf_{\theta \in \mathbb{R}^m} \phi_{\gamma,\rho_v}(x, y, \theta)$$

over $X \times \mathbb{R}^m$ by leveraging (Rockafellar, 1974, Theorem 1). $\qquad\square$

Next we develop a calculus for the Moreau Envelope $v_\gamma(x, y)$, providing formulas for its gradient. These results immediately give insights into the proposed algorithm. The proof closely follows that of Theorem 2.2 in Gao et al. (2023).

**Lemma A.2.** *Under Assumption of Lemma A.1, suppose that the gradient $\nabla_x g(x, y)$ exists and is continuous on $X \times Y$. The for $\gamma \in (0, \frac{1}{2\rho_{\varphi_2}})$, $S_\gamma(x, y) = \{\theta_\gamma^*(x, y)\}$ is a singleton. Furthermore,*

$$\nabla v_\gamma(x, y) = \left(\nabla_x f(x, \theta_\gamma^*(x, y)) + \nabla_x g(x, \theta_\gamma^*(x, y)), \left(y - \theta_\gamma^*(x, y)\right)/\gamma\right). \qquad (24)$$

*Proof.* Considering $\gamma \in (0, \frac{1}{2\rho_{\varphi_2}})$ and the weakly convexity of $\varphi(x, y)$, the function $\varphi(x, \theta) + \frac{1}{2\gamma}\|\theta - y\|^2 + \delta_Y(\theta)$ is shown to be $(\frac{1}{\gamma} - \frac{1}{\rho_{\varphi_2}})$-strongly convex with respect to $\theta$. Consequently, $S_\gamma(x, y) = \{\theta_*^*(x, y)\}$ is a singleton.

Further, for $\gamma \in (0, \frac{1}{2\rho_{\varphi_2}})$, we have $\rho_{v_1} \geq \rho_{\varphi_1}$ and $\rho_{v_2} \geq \frac{1}{\gamma}$. Leveraging Lemma A.1 and its subsequent proof, the function $v_\gamma(x, y) + \frac{\rho_{v_1}}{2}\|x\|^2 + \frac{\rho_{v_2}}{2}\|y\|^2$ is established as convex, and for any $(x, y) \in X \times Y$, the following holds

$$v_\gamma(x, y) + \frac{\rho_{v_1}}{2}\|x\|^2 + \frac{\rho_{v_2}}{2}\|y\|^2 = \inf_{\theta \in Y}\left\{\varphi(x, \theta) + \frac{\rho_{v_1}}{2}\|x\|^2 + \frac{\rho_{v_2}}{2}\|y\|^2 + \frac{1}{2\gamma}\|\theta - y\|^2\right\},$$

where $\varphi(x, \theta) + \frac{\rho_{v_1}}{2}\|x\|^2 + \frac{\rho_{v_2}}{2}\|y\|^2 + \frac{1}{2\gamma}\|\theta - y\|^2$ is convex with respect to $(x, y, \theta)$. By applying (Ye et al., 2023, Theorem 3) and exploiting the continuously differentiable property of $g(x, y)$ with respect to $x$, the desired formulas are derived. $\qquad\square$

## A.6 Auxiliary lemmas

In this section, we present auxiliary lemmas crucial for the non-asymptotic convergence analysis.

**Lemma A.3.** *Let $\gamma \in (0, \frac{1}{2\rho_{\varphi_2}})$, $(\bar{x}, \bar{y}) \in X \times \mathbb{R}^m$. Then for any $\rho_{v_1} \geq \rho_{\varphi_1}$, $\rho_{v_2} \geq \frac{1}{\gamma}$ and $(x, y)$ on $X \times \mathbb{R}^m$, the following inequality holds:*

$$-v_\gamma(x, y) \leq -v_\gamma(\bar{x}, \bar{y}) - \langle\nabla v_\gamma(\bar{x}, \bar{y}), (x, y) - (\bar{x}, \bar{y})\rangle + \frac{\rho_{v_1}}{2}\|x - \bar{x}\|^2 + \frac{\rho_{v_2}}{2}\|y - \bar{y}\|^2. \qquad (25)$$

*Proof.* According to Lemma A.1, $v_\gamma(x, y) + \frac{\rho_{v_1}}{2}\|x\|^2 + \frac{\rho_{v_2}}{2}\|y\|^2$ is convex on $X \times \mathbb{R}^m$. As a result, for any $(x, y)$ on $X \times \mathbb{R}^m$,

$$v_\gamma(x, y) + \frac{\rho_{v_1}}{2}\|x\|^2 + \frac{\rho_{v_2}}{2}\|y\|^2$$
$$\geq v_\gamma(\bar{x}, \bar{y}) + \frac{\rho_{v_1}}{2}\|\bar{x}\|^2 + \frac{\rho_{v_2}}{2}\|\bar{y}\|^2 + \langle\nabla v_\gamma(\bar{x}, \bar{y}) + (\rho_{v_1}\bar{x}, \rho_{v_2}\bar{y}), (x, y) - (\bar{x}, \bar{y})\rangle.$$

Consequently, the conclusion follows directly. $\qquad\square$

**Lemma A.4.** *For any $0 < s < 1/\rho_{g_2}$, and $\theta, \theta' \in \mathbb{R}^m$, the following inequality is satisfied:*

$$\|\text{Prox}_{s\tilde{g}(x,\cdot)}(\theta) - \text{Prox}_{s\tilde{g}(x,\cdot)}(\theta')\| \leq 1/(1 - s\rho_{g_2})\|\theta - \theta'\|. \qquad (26)$$

*Proof.* Let us denote $\text{Prox}_{s\tilde{g}(x,\cdot)}(\theta)$ and $\text{Prox}_{s\tilde{g}(x,\cdot)}(\theta')$ by $\theta^+$ and $\theta'^+$, respectively. From the definitions, we have

$$0 \in \partial_y \tilde{g}(x, \theta^+) + \frac{1}{s}(\theta^+ - \theta),$$

and

$$0 \in \partial_y \tilde{g}(x, \theta'^+) + \frac{1}{s}(\theta'^+ - \theta').$$

Given the $\rho_{g_2}$-weakly convexity of $\tilde{g}(x, \cdot)$, it implies

$$\left\langle -\frac{1}{s}(\theta^+ - \theta) + \frac{1}{s}(\theta'^+ - \theta'), \theta^+ - \theta'^+ \right\rangle \geq -\rho_{g_2} \|\theta^+ - \theta'^+\|^2.$$

From the above, the desired conclusion follows directly. $\qquad\square$

**Lemma A.5.** *Let $\gamma \in (0, \frac{1}{\rho_{f_2} + 2\rho_{g_2}})$. Then, there exists $L_\theta > 0$ such that for any $(x, y), (x', y') \in X \times \mathbb{R}^m$, the following inequality holds:*

$$\|\theta_\gamma^*(x, y) - \theta_\gamma^*(x', y')\| \leq L_\theta \|(x, y) - (x', y')\|. \tag{27}$$

*Proof.* Given that $\theta_\gamma^*(x, y)$ is optimal for the convex optimization problem $\min_{\theta \in Y} \varphi(x, \theta) + \frac{1}{2\gamma}\|\theta - y\|^2$, we have

$$0 \in \nabla_y f(x, \theta_\gamma^*(x, y)) + \partial_y g(x, \theta_\gamma^*(x, y)) + (\theta_\gamma^*(x, y) - y)/\gamma + \mathcal{N}_Y(\theta_\gamma^*(x, y)),$$
$$0 \in \nabla_y f(x', \theta_\gamma^*(x', y')) + \partial_y g(x', \theta_\gamma^*(x', y')) + (\theta_\gamma^*(x', y') - y')/\gamma + \mathcal{N}_Y(\theta_\gamma^*(x', y')).$$

Due to the $\rho_{g_2}$-weakly convexity of $\tilde{g}(x, y) := g(x, y) + \delta_Y(y)$ with respect to $y$, we obtain

$$\theta_\gamma^*(x, y) = \text{Prox}_{s\tilde{g}(x, \cdot)} \left( \theta_\gamma^*(x, y) - s \left( \nabla_y f(x, \theta_\gamma^*(x, y)) + (\theta_\gamma^*(x, y) - y)/\gamma \right) \right),$$
$$\theta_\gamma^*(x', y') = \text{Prox}_{s\tilde{g}(x', \cdot)} \left( \theta_\gamma^*(x', y') - s \left( \nabla_y f(x', \theta_\gamma^*(x', y')) + (\theta_\gamma^*(x', y') - y')/\gamma \right) \right), \tag{28}$$

when $0 < s < 1/\rho_{g_2}$. Consequently, we deduce that

$$\|\theta_\gamma^*(x, y) - \theta_\gamma^*(x', y')\|$$
$$= \big\| \text{Prox}_{s\tilde{g}(x, \cdot)} \left( \theta_\gamma^*(x, y) - s \left( \nabla_y f(x, \theta_\gamma^*(x, y)) + (\theta_\gamma^*(x, y) - y)/\gamma \right) \right)$$
$$\quad - \text{Prox}_{s\tilde{g}(x', \cdot)} \left( \theta_\gamma^*(x', y') - s \left( \nabla_y f(x', \theta_\gamma^*(x', y')) + (\theta_\gamma^*(x', y') - y')/\gamma \right) \right) \big\|$$
$$\leq \big\| \text{Prox}_{s\tilde{g}(x, \cdot)} \left( \theta_\gamma^*(x, y) - s \left( \nabla_y f(x, \theta_\gamma^*(x, y)) + (\theta_\gamma^*(x, y) - y)/\gamma \right) \right)$$
$$\quad - \text{Prox}_{s\tilde{g}(x, \cdot)} \left( \theta_\gamma^*(x', y') - s \left( \nabla_y f(x', \theta_\gamma^*(x', y')) + (\theta_\gamma^*(x', y') - y')/\gamma \right) \right) \big\|$$
$$+ \big\| \text{Prox}_{s\tilde{g}(x, \cdot)} \left( \theta_\gamma^*(x', y') - s \left( \nabla_y f(x', \theta_\gamma^*(x', y')) + (\theta_\gamma^*(x', y') - y')/\gamma \right) \right)$$
$$\quad - \text{Prox}_{s\tilde{g}(x', \cdot)} \left( \theta_\gamma^*(x', y') - s \left( \nabla_y f(x', \theta_\gamma^*(x', y')) + (\theta_\gamma^*(x', y') - y')/\gamma \right) \right) \big\| \tag{29}$$
$$\leq \big\| \text{Prox}_{s\tilde{g}(x, \cdot)} \left( \theta_\gamma^*(x, y) - s \left( \nabla_y f(x, \theta_\gamma^*(x, y)) + (\theta_\gamma^*(x, y) - y)/\gamma \right) \right)$$
$$\quad - \text{Prox}_{s\tilde{g}(x, \cdot)} \left( \theta_\gamma^*(x', y') - s \left( \nabla_y f(x, \theta_\gamma^*(x', y')) + (\theta_\gamma^*(x', y') - y)/\gamma \right) \right) \big\|$$
$$+ \big\| \text{Prox}_{s\tilde{g}(x, \cdot)} \left( \theta_\gamma^*(x', y') - s \left( \nabla_y f(x, \theta_\gamma^*(x', y')) + (\theta_\gamma^*(x', y') - y)/\gamma \right) \right)$$
$$\quad - \text{Prox}_{s\tilde{g}(x, \cdot)} \left( \theta_\gamma^*(x', y') - s \left( \nabla_y f(x', \theta_\gamma^*(x', y')) + (\theta_\gamma^*(x', y') - y')/\gamma \right) \right) \big\|$$
$$+ L_{\tilde{g}} \|x - x'\|,$$

where the second inequality is a consequence of Assumption 3.2(ii), which states that $\|\text{Prox}_{s\tilde{g}(x, \cdot)}(\theta) - \text{Prox}_{s\tilde{g}(x', \cdot)}(\theta)\| \leq L_{\tilde{g}}\|x - x'\|$ for any $\theta \in Y$ and $s \in (0, \bar{s}]$. Invoking Lemma A.4, for $0 < s < 1/\rho_{g_2}$, we derive

$$\|\text{Prox}_{s\tilde{g}(x, \cdot)}(\theta) - \text{Prox}_{s\tilde{g}(x, \cdot)}(\theta')\| \leq 1/(1 - s\rho_{g_2})\|\theta - \theta'\| \quad \forall \theta, \theta' \in \mathbb{R}^m. \tag{30}$$

Given that $f(x, \theta) + \frac{1}{2\gamma}\|\theta - y\|^2$ is $(\frac{1}{\gamma} - \rho_{f_2})$-strongly convex with respect to $\theta$ on $Y$, we have

$$\left\langle \nabla_y f(x, \theta_\gamma^*(x, y)) + (\theta_\gamma^*(x, y) - y)/\gamma - \nabla_y f(x, \theta_\gamma^*(x', y')) - (\theta_\gamma^*(x', y') - y)/\gamma, \theta_\gamma^*(x, y) - \theta_\gamma^*(x', y') \right\rangle$$
$$\geq \left( \frac{1}{\gamma} - \rho_{f_2} \right) \|\theta_\gamma^*(x, y) - \theta_\gamma^*(x', y')\|^2,$$

which implies that when $0 < s \le (1/\gamma - \rho_{f_2})/(L_f + 1/\gamma)^2$,

$$
\begin{aligned}
&\big\| \theta_\gamma^*(x,y) - s\left(\nabla_y f(x, \theta_\gamma^*(x,y)) + (\theta_\gamma^*(x,y) - y)/\gamma\right) - \theta_\gamma^*(x',y') \\
&\quad + s\left(\nabla_y f(x, \theta_\gamma^*(x',y')) + (\theta_\gamma^*(x',y') - y)/\gamma\right) \big\|^2 \\
&\le \left[1 - 2s\left(1/\gamma - \rho_{f_2}\right) + s^2(L_f + 1/\gamma)^2\right] \|\theta_\gamma^*(x,y) - \theta_\gamma^*(x',y')\|^2 \\
&\le \left[1 - s\left(1/\gamma - \rho_{f_2}\right)\right] \|\theta_\gamma^*(x,y) - \theta_\gamma^*(x',y')\|^2.
\end{aligned}
$$

Combining this with (30), we infer that

$$
\begin{aligned}
&\big\| \mathrm{Prox}_{s\tilde{g}(x,\cdot)}\left(\theta_\gamma^*(x,y) - s\left(\nabla_y f(x, \theta_\gamma^*(x,y)) + (\theta_\gamma^*(x,y) - y)/\gamma\right)\right) \\
&\quad - \mathrm{Prox}_{s\tilde{g}(x,\cdot)}\left(\theta_\gamma^*(x',y') - s\left(\nabla_y f(x, \theta_\gamma^*(x',y')) + (\theta_\gamma^*(x',y') - y)/\gamma\right)\right) \big\| \\
&\le 1/(1 - s\rho_{g_2}) \big\| \theta_\gamma^*(x,y) - s\left(\nabla_y f(x, \theta_\gamma^*(x,y)) + (\theta_\gamma^*(x,y) - y)/\gamma\right) \\
&\qquad\qquad - \theta_\gamma^*(x',y') + s\left(\nabla_y f(x, \theta_\gamma^*(x',y')) + (\theta_\gamma^*(x',y') - y)/\gamma\right) \big\| \\
&\le \sqrt{1 - s\left(1/\gamma - \rho_{f_2}\right)}/(1 - s\rho_{g_2}) \|\theta_\gamma^*(x,y) - \theta_\gamma^*(x',y')\|.
\end{aligned}
\tag{31}
$$

Next, utilizing Lemma A.4, for $0 < s < 1/\rho_{g_2}$, it follows that

$$
\begin{aligned}
&\big\| \mathrm{Prox}_{s\tilde{g}(x,\cdot)}\left(\theta_\gamma^*(x',y') - s\left(\nabla_y f(x, \theta_\gamma^*(x',y')) + (\theta_\gamma^*(x',y') - y)/\gamma\right)\right) \\
&\quad - \mathrm{Prox}_{s\tilde{g}(x,\cdot)}\left(\theta_\gamma^*(x',y') - s\left(\nabla_y f(x', \theta_\gamma^*(x',y')) + (\theta_\gamma^*(x',y') - y')/\gamma\right)\right) \big\| \\
&\le 1/(1 - s\rho_{g_2}) \big\| \theta_\gamma^*(x',y') - s\left(\nabla_y f(x, \theta_\gamma^*(x',y')) + (\theta_\gamma^*(x',y') - y)/\gamma\right) \\
&\qquad\qquad - \theta_\gamma^*(x',y') + s\left(\nabla_y f(x', \theta_\gamma^*(x',y')) + (\theta_\gamma^*(x',y') - y')/\gamma\right) \big\| \\
&\le s/(1 - s\rho_{g_2})\left(\|\nabla_y f(x, \theta_\gamma^*(x',y')) - \nabla_y f(x', \theta_\gamma^*(x',y'))\| + \|y - y'\|/\gamma\right) \\
&\le s/(1 - s\rho_{g_2})\left(L_f \|x - x'\| + \frac{1}{\gamma}\|y - y'\|\right).
\end{aligned}
\tag{32}
$$

From estimates (29), (31) and (32), we deduce that, for any $s > 0$ satisfying $s \le (1/\gamma - \rho_{f_2})/(L_f + 1/\gamma)^2$, $s \le \bar{s}$ and $s < 1/\rho_{g_2}$, the following condition holds

$$
\begin{aligned}
\|\theta_\gamma^*(x,y) - \theta_\gamma^*(x',y')\| &\le \sqrt{1 - s\left(1/\gamma - \rho_{f_2}\right)}/(1 - s\rho_{g_2}) \|\theta_\gamma^*(x,y) - \theta_\gamma^*(x',y')\| \\
&\quad + s/(1 - s\rho_{g_2})\left(L_f \|x - x'\| + \frac{1}{\gamma}\|y - y'\|\right) + L_{\tilde{g}}\|x - x'\|.
\end{aligned}
\tag{33}
$$

Given that $\gamma < \frac{1}{\rho_{f_2} + 2\rho_{g_2}}$, it can be inferred that $1/\gamma - \rho_{f_2} > 2\rho_{g_2}$. This implies $1 - 2s\rho_{g_2} > 1 - s(1/\gamma - \rho_{f_2})$, leading to $1 - s(1/\gamma - \rho_{f_2}) < (1 - s\rho_{g_2})^2$. Consequently, we deduce $\sqrt{1 - s\left(1/\gamma - \rho_{f_2}\right)}/(1 - s\rho_{g_2}) < 1$. From these derivations, the desired conclusion is evident. $\quad\square$

**Lemma A.6.** *Suppose $\gamma \in (0, \frac{1}{\rho_{f_2} + 2\rho_{g_2}})$ and $\eta_k \in (0, (1/\gamma - \rho_{f_2})/(L_f + 1/\gamma)^2] \cap (0, 1/\rho_{g_2})$, the sequence of $(x^k, y^k, \theta^k)$ generated by Algorithm 1 satisfies*

$$
\|\theta^{k+1} - \theta_\gamma^*(x^k, y^k)\| \le \sigma_k \|\theta^k - \theta_\gamma^*(x^k, y^k)\|,
\tag{34}
$$

*where $\sigma_k := \sqrt{1 - \eta_k\left(1/\gamma - \rho_{f_2}\right)}/(1 - \eta_k\rho_{g_2}) < 1$.*

*Proof.* Recalling (28) from Lemma A.5 that when $\eta_k < 1/\rho_{g_2}$,

$$
\theta_\gamma^*(x^k, y^k) = \mathrm{Prox}_{\eta_k\tilde{g}(x^k,\cdot)}\left(\theta_\gamma^*(x^k, y^k) - \eta_k\left(\nabla_y f(x^k, \theta_\gamma^*(x,y)) + (\theta_\gamma^*(x^k, y^k) - y^k)/\gamma\right)\right).
$$

Considering the update rule for $\theta^{k+1}$ as defined in (8) and using arguments analogous to those in the derivation of (31) from Lemma A.5, when $\eta_k \le (1/\gamma - \rho_{f_2})/(L_f + 1/\gamma)^2$, it follows

$$
\|\theta^{k+1} - \theta_\gamma^*(x^k, y^k)\| \le \sigma_k \|\theta^k - \theta_\gamma^*(x^k, y^k)\|,
$$

where $\sigma_k := \sqrt{1 - \eta_k\left(1/\gamma - \rho_{f_2}\right)}/(1 - \eta_k\rho_{g_2})$. Notably, $\sigma_k < 1$ is a consequence of $\gamma < \frac{1}{\rho_{f_2} + 2\rho_{g_2}}$. $\quad\square$

As previously highlighted, the update of variables $(x, y)$ in (9) can be interpreted as inexact alternating proximal gradient from $(x^k, y^k)$ on $\min_{(x,y) \in X \times Y} \phi_{c_k}(x, y)$, in which $\phi_{c_k}$ is defined in (12) as

$$\phi_{c_k}(x, y) := \frac{1}{c_k} \big(F(x, y) - \underline{F}\big) + f(x, y) + g(x, y) - v_\gamma(x, y).$$

The subsequent lemma illustrates that the function $\phi_{c_k}(x, y)$ exhibits a monotonic decreasing behavior with errors at each iteration.

**Lemma A.7.** *Under Assumptions 3.1 and 3.2, suppose $\gamma \in (0, \frac{1}{2\rho_{f_2} + 2\rho_{g_2}})$ and $\beta_k < 1/\rho_{g_2}$, the sequence of $(x^k, y^k, \theta^k)$ generated by Algorithm 1 satisfies*

$$\begin{aligned}
\phi_{c_k}(x^{k+1}, y^{k+1}) \leq{}& \phi_{c_k}(x^k, y^k) - \left(\frac{1}{2\alpha_k} - \frac{L_{\phi_k}}{2} - \frac{\beta_k L_\theta^2}{\gamma^2}\right) \|x^{k+1} - x^k\|^2 \\
&- \left(\frac{1}{2\beta_k} - \frac{\rho_{g_2}}{2} - \frac{L_{\phi_k}}{2}\right) \|y^{k+1} - y^k\|^2 \\
&+ \left(\frac{\alpha_k}{2}(L_f + L_g)^2 + \frac{\beta_k}{\gamma^2}\right) \left\|\theta^{k+1} - \theta_\gamma^*(x^k, y^k)\right\|^2,
\end{aligned} \tag{35}$$

*where $L_{\phi_k} := L_F/c_k + L_f + L_g + \max\{\rho_{\varphi_1}, 1/\gamma\}$.*

*Proof.* Under the conditions of Assumptions 3.1 and 3.2(i), the functions $F$ and $f$ exhibit $L_F$- and $L_f$-smooth on $X \times Y$, respectively. Further, according to Assumption 3.2(ii), the function $g(\cdot, y^k)$ is $L_g$-smooth on $X$. Leveraging these assumptions and invoking Lemma A.3, we deduce

$$\phi_{c_k}(x^{k+1}, y^k) \leq \phi_{c_k}(x^k, y^k) + \langle \nabla_x \phi_{c_k}(x^k, y^k), x^{k+1} - x^k \rangle + \frac{L_{\phi_k}}{2} \|x^{k+1} - x^k\|^2, \tag{36}$$

with $L_{\phi_k} := L_F/c_k + L_f + L_g + \max\{\rho_{\varphi_1}, 1/\gamma\}$. Considering the update rule for the variable $x$ as defined in (9) and leveraging the property of the projection operator $\mathrm{Proj}_X$, it follows that

$$\langle x^k - \alpha_k d_x^k - x^{k+1}, x^k - x^{k+1} \rangle \leq 0,$$

leading to

$$\langle d_x^k, x^{k+1} - x^k \rangle \leq -\frac{1}{\alpha_k} \|x^{k+1} - x^k\|^2.$$

Combining this inequality with (36), if can be deduced that

$$\begin{aligned}
\phi_{c_k}(x^{k+1}, y^k) \leq{}& \phi_{c_k}(x^k, y^k) - \left(\frac{1}{\alpha_k} - \frac{L_{\phi_k}}{2}\right) \|x^{k+1} - x^k\|^2 \\
&+ \langle \nabla_x \phi_{c_k}(x^k, y^k) - d_x^k, x^{k+1} - x^k \rangle.
\end{aligned} \tag{37}$$

Given the expression for $\nabla v_\gamma(x, y)$ as derived in Lemma A.2 and the definition of $d_x^k$ provided in (10), we obtain

$$\begin{aligned}
&\left\| \nabla_x \phi_{c_k}(x^k, y^k) - d_x^k \right\|^2 \\
={}& \left\| \nabla_x f(x^k, \theta_\gamma^*(x^k, y^k)) + \nabla_x g(x^k, \theta_\gamma^*(x^k, y^k)) - \nabla_x f(x^k, \theta^{k+1}) - \nabla_x g(x^k, \theta^{k+1}) \right\|^2 \quad (38) \\
\leq{}& (L_f + L_g)^2 \left\| \theta^{k+1} - \theta_\gamma^*(x^k, y^k) \right\|^2.
\end{aligned}$$

This yields

$$\begin{aligned}
&\langle \nabla_x \phi_{c_k}(x^k, y^k) - d_x^k, x^{k+1} - x^k \rangle \\
\leq{}& \frac{\alpha_k}{2}(L_f + L_g)^2 \left\| \theta^{k+1} - \theta_\gamma^*(x^k, y^k) \right\|^2 + \frac{1}{2\alpha_k} \|x^{k+1} - x^k\|^2,
\end{aligned}$$

which combining with (37) leads to

$$\begin{aligned}
\phi_{c_k}(x^{k+1}, y^k) \leq{}& \phi_{c_k}(x^k, y^k) - \left(\frac{1}{2\alpha_k} - \frac{L_{\phi_k}}{2}\right) \|x^{k+1} - x^k\|^2 \\
&+ \frac{\alpha_k}{2}(L_f + L_g)^2 \left\| \theta^{k+1} - \theta_\gamma^*(x^k, y^k) \right\|^2.
\end{aligned} \tag{39}$$

Considering the update rule for variable $y$ given by (9), and the $\rho_{g_2}$-weakly convex property of $g(x^{k+1}, \cdot)$ over $Y$, it follows that for $\beta_k < 1/\rho_{g_2}$,

$$\langle d_y^k, y^{k+1} - y^k \rangle + g(x^{k+1}, y^{k+1}) + \left( \frac{1}{\beta_k} - \frac{\rho_{g_2}}{2} \right) \|y^{k+1} - y^k\|^2 \le g(x^{k+1}, y^k). \quad (40)$$

Under Assumptions 3.1 and 3.2(i), where $F$ and $f$ are $L_F$- and $l_f$-smooth on $X \times Y$, respectively, and invoking Lemma A.3, we deduce

$$\phi_{c_k}(x^{k+1}, y^{k+1}) - g(x^{k+1}, y^{k+1})$$
$$\le \phi_{c_k}(x^{k+1}, y^k) - g(x^{k+1}, y^k) + \langle \nabla_y (\phi_{c_k} - g)(x^{k+1}, y^k), y^{k+1} - y^k \rangle + \frac{L_{\phi_k}}{2} \|y^{k+1} - y^k\|^2. \quad (41)$$

Combining this inequality with (40), we obtain

$$\phi_{c_k}(x^{k+1}, y^{k+1})$$
$$\le \phi_{c_k}(x^{k+1}, y^k) - \left( \frac{1}{\beta_k} - \frac{\rho_{g_2}}{2} - \frac{L_{\phi_k}}{2} \right) \|y^{k+1} - y^k\|^2 \quad (42)$$
$$+ \langle \nabla_y (\phi_{c_k} - g)(x^{k+1}, y^k) - d_y^k, y^{k+1} - y^k \rangle.$$

Given the expression for $\nabla v_\gamma(x, y)$ as derived in Lemma A.2 and the definition of $d_y^k$ from (10), we deduce

$$\left\| \nabla_y (\phi_{c_k} - g)(x^{k+1}, y^k) - d_y^k \right\|^2 = \left\| (y^k - \theta_\gamma^*(x^{k+1}, y^k))/\gamma - (y^k - \theta^{k+1})/\gamma \right\|^2$$
$$= \frac{1}{\gamma^2} \left\| \theta^{k+1} - \theta_\gamma^*(x^{k+1}, y^k) \right\|^2, \quad (43)$$

and thus

$$\langle \nabla_y (\phi_{c_k} - g)(x^{k+1}, y^k) - d_y^k, y^{k+1} - y^k \rangle \le \frac{\beta_k}{2\gamma^2} \left\| \theta^{k+1} - \theta_\gamma^*(x^{k+1}, y^k) \right\|^2 + \frac{1}{2\beta_k} \|y^{k+1} - y^k\|^2.$$

Consequently, we have from (42) that

$$\phi_{c_k}(x^{k+1}, y^{k+1})$$
$$\le \phi_{c_k}(x^{k+1}, y^k) - \left( \frac{1}{2\beta_k} - \frac{\rho_{g_2}}{2} - \frac{L_{\phi_k}}{2} \right) \|y^{k+1} - y^k\|^2 + \frac{\beta_k}{2\gamma^2} \left\| \theta^{k+1} - \theta_\gamma^*(x^{k+1}, y^k) \right\|^2$$
$$\le \phi_{c_k}(x^{k+1}, y^k) - \left( \frac{1}{2\beta_k} - \frac{\rho_{g_2}}{2} - \frac{L_{\phi_k}}{2} \right) \|y^{k+1} - y^k\|^2 + \frac{\beta_k}{\gamma^2} \left\| \theta^{k+1} - \theta_\gamma^*(x^k, y^k) \right\|^2 \quad (44)$$
$$+ \frac{\beta_k L_\theta^2}{\gamma^2} \left\| x^{k+1} - x^k \right\|^2,$$

where the last inequality follows from Lemma A.5. The conclusion follows by combining this with (39). $\qquad \square$

## A.7 PROOF OF LEMMA 3.1

Leveraging the auxiliary lemmas from the preceding section, we demonstrate the decreasing property of the merit function $V_k$.

**Lemma A.8.** *Under Assumptions 3.1 and 3.2, suppose $\gamma \in (0, \frac{1}{2\rho_{f_2} + 2\rho_{g_2}})$, $c_{k+1} \ge c_k$ and $\eta_k \in [\underline{\eta}, (1/\gamma - \rho_{f_2})/(L_f + 1/\gamma)^2] \cap [\underline{\eta}, 1/\rho_{g_2})$ with $\underline{\eta} > 0$, then there exists $c_\alpha, c_\beta, c_\theta > 0$ such that when $0 < \alpha_k \le c_\alpha$ and $0 < \beta_k \le c_\beta$, the sequence of $(x^k, y^k, \theta^k)$ generated by Algorithm 1 satisfies*

$$V_{k+1} - V_k \le -\frac{1}{4\alpha_k} \|x^{k+1} - x^k\|^2 - \frac{1}{4\beta_k} \|y^{k+1} - y^k\|^2 - c_\theta \left\| \theta^k - \theta_\gamma^*(x^k, y^k) \right\|^2, \quad (45)$$

*where $c_\theta = \frac{1}{2} \left( \frac{\underline{\eta} \rho_{g_2}}{1 - \underline{\eta} \rho_{g_2}} \right)^2 ((L_f + L_g)^2 + 1/\gamma^2)$.*

*Proof.* Let us first recall (35) from Lemma A.7, which states that

$$\phi_{c_k}(x^{k+1}, y^{k+1}) \leq \phi_{c_k}(x^k, y^k) - \left(\frac{1}{2\alpha_k} - \frac{L_{\phi_k}}{2} - \frac{\beta_k L_\theta^2}{\gamma^2}\right) \|x^{k+1} - x^k\|^2$$

$$- \left(\frac{1}{2\beta_k} - \frac{\rho_{g_2}}{2} - \frac{L_{\phi_k}}{2}\right) \|y^{k+1} - y^k\|^2 \tag{46}$$

$$+ \left(\frac{\alpha_k}{2}(L_f + L_g)^2 + \frac{\beta_k}{\gamma^2}\right) \left\|\theta^{k+1} - \theta_\gamma^*(x^k, y^k)\right\|^2,$$

when $\beta_k < 1/\rho_{g_2}$. Since $c_{k+1} \geq c_k$, we can infer that $(F(x^{k+1}, y^{k+1}) - \underline{F})/c_{k+1} \leq (F(x^{k+1}, y^{k+1}) - \underline{F})/c_k$. Combining this with (46) leads to

$$V_{k+1} - V_k = \phi_{c_{k+1}}(x^{k+1}, y^{k+1}) - \phi_{c_k}(x^k, y^k)$$

$$+ \left((L_f + L_g)^2 + 1/\gamma^2\right) \left\|\theta^{k+1} - \theta_\gamma^*(x^{k+1}, y^{k+1})\right\|^2$$

$$- \left((L_f + L_g)^2 + 1/\gamma^2\right) \left\|\theta^k - \theta_\gamma^*(x^k, y^k)\right\|^2$$

$$\leq \phi_{c_k}(x^{k+1}, y^{k+1}) - \phi_{c_k}(x^k, y^k)$$

$$+ \left((L_f + L_g)^2 + 1/\gamma^2\right) \left\|\theta^{k+1} - \theta_\gamma^*(x^{k+1}, y^{k+1})\right\|^2$$

$$- \left((L_f + L_g)^2 + 1/\gamma^2\right) \left\|\theta^k - \theta_\gamma^*(x^k, y^k)\right\|^2$$

$$\leq - \left(\frac{1}{2\alpha_k} - \frac{L_{\phi_k}}{2} - \frac{\beta_k L_\theta^2}{\gamma^2}\right) \|x^{k+1} - x^k\|^2 \tag{47}$$

$$- \left(\frac{1}{2\beta_k} - \frac{\rho_{g_2}}{2} - \frac{L_{\phi_k}}{2}\right) \|y^{k+1} - y^k\|^2$$

$$+ \left((L_f + L_g)^2 + 1/\gamma^2\right) \left\|\theta^{k+1} - \theta_\gamma^*(x^{k+1}, y^{k+1})\right\|^2$$

$$- \left((L_f + L_g)^2 + 1/\gamma^2\right) \left\|\theta^k - \theta_\gamma^*(x^k, y^k)\right\|^2$$

$$+ \left(\frac{\alpha_k}{2}(L_f + L_g)^2 + \frac{\beta_k}{\gamma^2}\right) \left\|\theta^{k+1} - \theta_\gamma^*(x^k, y^k)\right\|^2.$$

We can demonstrate that

$$\left\|\theta^{k+1} - \theta_\gamma^*(x^{k+1}, y^{k+1})\right\|^2 - \left\|\theta^k - \theta_\gamma^*(x^k, y^k)\right\|^2 + \frac{\alpha_k}{2} \left\|\theta^{k+1} - \theta_\gamma^*(x^k, y^k)\right\|^2$$

$$\leq (1 + \epsilon_k + \frac{\alpha_k}{2}) \left\|\theta^{k+1} - \theta_\gamma^*(x^k, y^k)\right\|^2 - \left\|\theta^k - \theta_\gamma^*(x^k, y^k)\right\|^2$$

$$+ (1 + \frac{1}{\epsilon_k})\|\theta_\gamma^*(x^{k+1}, y^{k+1}) - \theta_\gamma^*(x^k, y^k)\|^2$$

$$\leq (1 + \epsilon_k + \frac{\alpha_k}{2})\sigma_k^2\|\theta^k - \theta_\gamma^*(x^k, y^k)\|^2 - \left\|\theta^k - \theta_\gamma^*(x^k, y^k)\right\|^2$$

$$+ (1 + \frac{1}{\epsilon_k})L_\theta^2 \left\|(x^{k+1}, y^{k+1}) - (x^k, y^k)\right\|^2,$$

for any $\epsilon_k > 0$, where the second inequality is a consequence of Lemmas A.5 and A.6. Since $\gamma < \frac{1}{\rho_{f_2} + 2\rho_{g_2}}$, we have $1 - 2\eta_k \rho_{g_2} > 1 - \eta_k(1/\gamma - \rho_{f_2})$, and thus $\sigma_k^2 = (1 - \eta_k(1/\gamma - \rho_{f_2}))/(1 - \eta_k \rho_{g_2})^2 \leq 1 - \left(\frac{\eta_k \rho_{g_2}}{1 - \eta_k \rho_{g_2}}\right)^2$. By setting $\epsilon_k = \frac{1}{4}\left(\frac{\eta_k \rho_{g_2}}{1 - \eta_k \rho_{g_2}}\right)^2$ in the above inequality, we deduce that when $\alpha_k \leq \frac{1}{2}\left(\frac{\eta_k \rho_{g_2}}{1 - \eta_k \rho_{g_2}}\right)^2$, it follows that

$$\left\|\theta^{k+1} - \theta_\gamma^*(x^{k+1}, y^{k+1})\right\|^2 - \left\|\theta^k - \theta_\gamma^*(x^k, y^k)\right\|^2 + \frac{\alpha_k}{2} \left\|\theta^{k+1} - \theta_\gamma^*(x^k, y^k)\right\|^2$$

$$\leq -\frac{1}{2}\left(\frac{\eta_k \rho_{g_2}}{1 - \eta_k \rho_{g_2}}\right)^2 \left\|\theta^k - \theta_\gamma^*(x^k, y^k)\right\|^2 \tag{48}$$

$$+ \left(1 + 4\left(\frac{1 - \eta_k \rho_{g_2}}{\eta_k \rho_{g_2}}\right)^2\right) L_\theta^2 \left\|(x^{k+1}, y^{k+1}) - (x^k, y^k)\right\|^2.$$

Similarly, we can show that, when $\beta_k \leq \frac{1}{4}\left(\frac{\eta_k \rho_{g_2}}{1 - \eta_k \rho_{g_2}}\right)^2$, it holds that

$$
\begin{aligned}
&\left\|\theta^{k+1} - \theta_\gamma^*(x^{k+1}, y^{k+1})\right\|^2 - \left\|\theta^k - \theta_\gamma^*(x^k, y^k)\right\|^2 + \beta_k \left\|\theta^{k+1} - \theta_\gamma^*(x^k, y^k)\right\|^2 \\
&\leq -\frac{1}{2}\left(\frac{\eta_k \rho_{g_2}}{1 - \eta_k \rho_{g_2}}\right)^2 \left\|\theta^k - \theta_\gamma^*(x^k, y^k)\right\|^2 \\
&\quad + \left(1 + 4\left(\frac{1 - \eta_k \rho_{g_2}}{\eta_k \rho_{g_2}}\right)^2\right) L_\theta^2 \left\|(x^{k+1}, y^{k+1}) - (x^k, y^k)\right\|^2.
\end{aligned}
\tag{49}
$$

Combining (47), (48) and (49), we have

$$
\begin{aligned}
&V_{k+1} - V_k \\
&\leq -\left[\frac{1}{2\alpha_k} - \frac{L_{\phi_k}}{2} - \frac{\beta_k L_\theta^2}{\gamma^2} - \left(1 + 4\left(\frac{1 - \eta_k \rho_{g_2}}{\eta_k \rho_{g_2}}\right)^2\right) L_\theta^2 \left((L_f + L_g)^2 + 1/\gamma^2\right)\right] \|x^{k+1} - x^k\|^2 \\
&\quad -\left[\frac{1}{2\beta_k} - \frac{\rho_{g_2}}{2} - \frac{L_{\phi_k}}{2} - \left(1 + 4\left(\frac{1 - \eta_k \rho_{g_2}}{\eta_k \rho_{g_2}}\right)^2\right) L_\theta^2 \left((L_f + L_g)^2 + 1/\gamma^2\right)\right] \|y^{k+1} - y^k\|^2 \\
&\quad -\frac{1}{2}\left(\frac{\eta_k \rho_{g_2}}{1 - \eta_k \rho_{g_2}}\right)^2 \left((L_f + L_g)^2 + 1/\gamma^2\right) \left\|\theta^k - \theta_\gamma^*(x^k, y^k)\right\|^2.
\end{aligned}
\tag{50}
$$

When $c_{k+1} \geq c_k$, $\eta_k \geq \underline{\eta} > 0$, $\alpha_k \leq \frac{1}{2}\left(\frac{\underline{\eta}\rho_{g_2}}{1 - \underline{\eta}\rho_{g_2}}\right)^2$ and $\beta_k \leq \frac{1}{4}\left(\frac{\underline{\eta}\rho_{g_2}}{1 - \underline{\eta}\rho_{g_2}}\right)^2$ and , it holds that, for any $k$, $\alpha_k \leq \frac{1}{2}\left(\frac{\eta_k \rho_{g_2}}{1 - \eta_k \rho_{g_2}}\right)^2$, $\beta_k \leq \frac{1}{4}\left(\frac{\eta_k \rho_{g_2}}{1 - \eta_k \rho_{g_2}}\right)^2$,

$$
\begin{aligned}
&\frac{L_{\phi_k}}{2} + \frac{\beta_k L_\theta^2}{\gamma^2} + \left(1 + 4\left(\frac{1 - \eta_k \rho_{g_2}}{\eta_k \rho_{g_2}}\right)^2\right) L_\theta^2 \left((L_f + L_g)^2 + 1/\gamma^2\right) \\
&\leq \frac{L_{\phi_0}}{2} + \frac{L_\theta^2}{4\gamma^2}\left(\frac{\underline{\eta}\rho_{g_2}}{1 - \underline{\eta}\rho_{g_2}}\right)^2 + \left(1 + 4\left(\frac{1 - \underline{\eta}\rho_{g_2}}{\underline{\eta}\rho_{g_2}}\right)^2\right) L_\theta^2 \left((L_f + L_g)^2 + 1/\gamma^2\right) =: C_\alpha,
\end{aligned}
\tag{51}
$$

and

$$
\begin{aligned}
&\frac{\rho_{g_2}}{2} + \frac{L_{\phi_k}}{2} + \left(1 + 4\left(\frac{1 - \eta_k \rho_{g_2}}{\eta_k \rho_{g_2}}\right)^2\right) L_\theta^2 \left((L_f + L_g)^2 + 1/\gamma^2\right) \\
&\leq \frac{\rho_{g_2}}{2} + \frac{L_{\phi_0}}{2} + \left(1 + 4\left(\frac{1 - \underline{\eta}\rho_{g_2}}{\underline{\eta}\rho_{g_2}}\right)^2\right) L_\theta^2 \left((L_f + L_g)^2 + 1/\gamma^2\right) =: C_\beta,
\end{aligned}
\tag{52}
$$

Consequently, since $\frac{1}{4C_\beta} < \frac{1}{2\rho_{g_2}}$, if $c_\alpha, c_\beta > 0$ satisfies

$$
c_\alpha \leq \min\left\{\frac{1}{2}\left(\frac{\underline{\eta}\rho_{g_2}}{1 - \underline{\eta}\rho_{g_2}}\right)^2, \frac{1}{4C_\alpha}\right\}, \quad c_\beta \leq \min\left\{\frac{1}{4}\left(\frac{\underline{\eta}\rho_{g_2}}{1 - \underline{\eta}\rho_{g_2}}\right)^2, \frac{1}{4C_\beta}\right\},
\tag{53}
$$

then, when $0 < \alpha_k \leq c_\alpha$ and $0 < \beta_k \leq c_\beta$, it holds that

$$
\frac{1}{2\alpha_k} - \frac{L_{\phi_k}}{2} - \frac{\beta_k L_\theta}{\gamma^2} - \left(1 + 4\left(\frac{1 - \eta_k \rho_{g_2}}{\eta_k \rho_{g_2}}\right)^2\right) L_\theta^2 \left((L_f + L_g)^2 + 1/\gamma^2\right) \geq \frac{1}{4\alpha_k},
$$

and

$$
\frac{1}{2\beta_k} - \frac{\rho_{g_2}}{2} - \frac{L_{\phi_k}}{2} - \left(1 + 4\left(\frac{1 - \eta_k \rho_{g_2}}{\eta_k \rho_{g_2}}\right)^2\right) L_\theta^2 \left((L_f + L_g)^2 + 1/\gamma^2\right) \geq \frac{1}{4\beta_k}.
$$

Consequently, the conclusion follows from (50). $\qquad\square$

A.8  PROOF OF THEOREM 3.1

By leveraging the monotonically decreasing property of the merit function $V_k$, we can establish the non-asymptotic convergence for the sequence $(x^k, y^k, \theta^k)$ generated by the proposed MEHA.

**Theorem A.2.** *Under Assumptions 3.1 and 3.2, suppose $\gamma \in (0, \frac{1}{2\rho_{f_2}+2\rho_{g_2}})$, $c_k = \underline{c}(k+1)^p$ with $p \in [0, 1/2)$, $\underline{c} > 0$ and $\eta_k \in [\underline{\eta}, (1/\gamma - \rho_{f_2})/(L_f + 1/\gamma)^2] \cap [\underline{\eta}, 1/\rho_{g_2})$ with $\underline{\eta} > 0$, then there exists $c_\alpha, c_\beta > 0$ such that when $\alpha_k \in (\underline{\alpha}, c_\alpha)$ and $\beta_k \in (\underline{\beta}, c_\beta)$ with $\underline{\alpha}, \underline{\beta} > 0$, the sequence of $(x^k, y^k, \theta^k)$ generated by Algorithm 1 satisfies*

$$\min_{0 \le k \le K} \left\| \theta^k - \theta_\gamma^*(x^k, y^k) \right\| = O\left( \frac{1}{K^{1/2}} \right),$$

*and*

$$\min_{0 \le k \le K} R_k(x^{k+1}, y^{k+1}) = O\left( \frac{1}{K^{(1-2p)/2}} \right).$$

*Furthermore, if there exists $M > 0$ such that $\psi_{c_k}(x^k, y^k) \le M$ for any $k$, the sequence of $(x^k, y^k)$ satisfies*

$$\varphi(x^K, y^K) - v_\gamma(x^K, y^K) = O\left( \frac{1}{K^p} \right).$$

*Proof.* First, Lemma 3.1 ensures the existence of $c_\alpha, c_\beta > 0$ for which (13) is valid under the conditions $\alpha_k \le c_\alpha, \beta_k \le c_\beta$. Upon telescoping (13) over the range $k = 0, 1, \ldots, K-1$, we derive

$$\sum_{k=0}^{K-1} \left( \frac{1}{4\alpha_k} \|x^{k+1} - x^k\|^2 + \frac{1}{4\beta_k} \|y^{k+1} - y^k\|^2 \right.$$
$$\left. + \frac{1}{2} \left( \frac{\underline{\eta}\rho_{g_2}}{1 - \underline{\eta}\rho_{g_2}} \right)^2 ((L_f + L_g)^2 + 1/\gamma^2) \left\| \theta^k - \theta_\gamma^*(x^k, y^k) \right\|^2 \right) \tag{54}$$
$$\le V_0 - V_K \le V_0,$$

where the last inequality is valid because $V_K$ is nonnegative. Thus, we have

$$\sum_{k=0}^{\infty} \left\| \theta^k - \theta_\gamma^*(x^k, y^k) \right\|^2 < \infty,$$

and then

$$\min_{0 \le k \le K} \left\| \theta^k - \theta_\gamma^*(x^k, y^k) \right\| = O\left( \frac{1}{K^{1/2}} \right).$$

According to the update rule of variables $(x, y)$ as defined in (9), we have that

$$0 \in c_k d_x^k + \mathcal{N}_X(x^{k+1}) + \frac{c_k}{\alpha_k}\left( x^{k+1} - x^k \right),$$
$$0 \in c_k d_y^k + c_k \partial_y g(x^{k+1}, y^{k+1}) + \mathcal{N}_Y(y^{k+1}) + \frac{c_k}{\beta_k}\left( y^{k+1} - y^k \right). \tag{55}$$

From the definitions of $d_x^k$ and $d_y^k$ provided in (10), and given $\nabla_x g(x^{k+1}, y^{k+1}) \times \partial_y g(x^{k+1}, y^{k+1}) \subseteq \partial g(x^{k+1}, y^{k+1})$, a result stemming from the weakly convexity of $g$ and its continuously differentiable property with respect to $x$ as outlined in Assumption 3.2(ii) and corroborated by (Gao et al., 2023, Proposition 2.1)), we deduce

$$(e_x^k, e_y^k) \in \nabla F(x^{k+1}, y^{k+1}) + c_k \left( \nabla f(x^{k+1}, y^{k+1}) + \partial g(x^{k+1}, y^{k+1}) - \nabla v_\gamma(x^{k+1}, y^{k+1}) \right)$$
$$+ \mathcal{N}_{X \times Y}(x^{k+1}, y^{k+1}),$$

with

$$e_x^k := \nabla_x \psi_{c_k}(x^{k+1}, y^{k+1}) - c_k d_x^k - \frac{c_k}{\alpha_k}\left( x^{k+1} - x^k \right),$$
$$e_y^k := \nabla_y (\psi_{c_k} - g)(x^{k+1}, y^{k+1}) - c_k d_y^k - \frac{c_k}{\beta_k}\left( y^{k+1} - y^k \right). \tag{56}$$

Next, we estimate $\|e_x^k\|$. We have

$$\|e_x^k\| \le \|\nabla_x \psi_{c_k}(x^{k+1}, y^{k+1}) - \nabla_x \psi_{c_k}(x^k, y^k)\| + \|\nabla_x \psi_{c_k}(x^k, y^k) - c_k d_x^k\| + \frac{c_k}{\alpha_k}\|x^{k+1} - x^k\|.$$

Considering the first term on the right hand side of the preceding inequality, and invoking Assumptions 3.1 and 3.2 alongside Lemma A.2, A.3, A.5, we establish the existence of $L_{\psi_1} > 0$ such that

$$\|\nabla_x \psi_{c_k}(x^{k+1}, y^{k+1}) - \nabla_x \psi_{c_k}(x^k, y^k)\| \le c_k L_{\psi_1}\|(x^{k+1}, y^{k+1}) - (x^k, y^k)\|.$$

Using (38) and Lemma A.6, we deduce

$$\|\nabla_x \psi_{c_k}(x^k, y^k) - c_k d_x^k\| = c_k\|\nabla_x \phi_{c_k}(x^k, y^k) - d_x^k\| \le c_k(L_f + L_g)\|\theta^k - \theta_\gamma^*(x^k, y^k)\|. \tag{57}$$

Hence, we have

$$\|e_x^k\| \le c_k L_{\psi_1}\|(x^{k+1}, y^{k+1}) - (x^k, y^k)\| + \frac{c_k}{\alpha_k}\|x^{k+1} - x^k\| + c_k(L_f + L_g)\|\theta^k - \theta_\gamma^*(x^k, y^k)\|.$$

For $\|e_y^k\|$, it follows that

$$\|e_y^k\| \le \|\nabla_y(\psi_{c_k} - g)(x^{k+1}, y^{k+1}) - \nabla_y(\psi_{c_k} - g)(x^{k+1}, y^k)\| + \frac{c_k}{\beta_k}\|y^{k+1} - y^k\|$$
$$+ \|\nabla_y(\psi_{c_k} - g)(x^{k+1}, y^k) - c_k d_y^k\|.$$

Analogously, invoking Assumptions 3.1 and 3.2 together with Lemmas A.2, A.3, and A.5, we have the existence of $L_{\psi_2} > 0$ such that

$$\|\nabla_y(\psi_{c_k} - g)(x^{k+1}, y^{k+1}) - \nabla_y(\psi_{c_k} - g)(x^{k+1}, y^k)\| \le c_k L_{\psi_2}\|y^{k+1} - y^k\|.$$

Using (43), Lemma A.5 and Lemma A.6, we obtain

$$\|\nabla_y(\psi_{c_k} - g)(x^{k+1}, y^k) - c_k d_y^k\| = c_k\|\nabla_y(\phi_{c_k} - g)(x^{k+1}, y^k) - d_y^k\|$$
$$\le \frac{c_k}{\gamma}\left(\|\theta^k - \theta_\gamma^*(x^k, y^k)\| + L_\theta\|x^{k+1} - x^k\|\right).$$

Therefore, we have

$$\|e_y^k\| \le c_k L_{\psi_2}\|y^{k+1} - y^k\| + \frac{c_k}{\beta_k}\|y^{k+1} - y^k\| + \frac{c_k}{\gamma}\left(\|\theta^k - \theta_\gamma^*(x^k, y^k)\| + L_\theta\|x^{k+1} - x^k\|\right).$$

With the estimations of $\|e_x^k\|$ and $\|e_y^k\|$, we obtain the existence of $L_\psi > 0$ such that

$$R_k(x^{k+1}, y^{k+1}) \le c_k L_\psi\|(x^{k+1}, y^{k+1}) - (x^k, y^k)\| + \left(\frac{c_k}{\alpha_k} + \frac{c_k L_\theta}{\gamma}\right)\|x^{k+1} - x^k\|$$
$$+ \frac{c_k}{\beta_k}\|y^{k+1} - y^k\| + c_k(L_f + L_g + \frac{1}{\gamma})\|\theta^k - \theta_\gamma^*(x^k, y^k)\|.$$

Employing the aforementioned inequality and given that $\alpha_k \ge \underline{\alpha}$ and $\beta_k \ge \underline{\beta}$ for some positive constants $\underline{\alpha}, \underline{\beta}$, we demonstrate the existence of $C_R > 0$ such that

$$\frac{1}{c_k^2} R_k(x^{k+1}, y^{k+1})^2$$
$$\le C_R\Big(\frac{1}{4\alpha_k}\|x^{k+1} - x^k\|^2 + \frac{1}{4\beta_k}\|y^{k+1} - y^k\|^2 \tag{58}$$
$$+ \frac{1}{2}\left(\frac{\eta\rho_{g_2}}{1 - \underline{\eta}\rho_{g_2}}\right)^2\left((L_f + L_g)^2 + 1/\gamma^2\right)\|\theta^k - \theta_\gamma^*(x^k, y^k)\|^2\Big).$$

Combining this with (54) implies that

$$\sum_{k=0}^\infty \frac{1}{c_k^2} R_k(x^{k+1}, y^{k+1})^2 < \infty. \tag{59}$$

Because $2p < 1$, it holds that

$$\sum_{k=0}^{K} \frac{1}{c_k^2} = \frac{1}{\underline{c}^2} \sum_{k=0}^{K} \left( \frac{1}{k+1} \right)^{2p} \geq \frac{1}{\underline{c}^2} \int_1^{K+2} \frac{1}{t^{2p}} dt \geq \frac{(K+2)^{1-2p} - 1}{(1-2p)\underline{c}^2},$$

and we can conclude from (59) that

$$\min_{0 \leq k \leq K} R_k(x^{k+1}, y^{k+1}) = O\left( \frac{1}{K^{(1-2p)/2}} \right).$$

Finally, since $\psi_{c_k}(x^k, y^k) \leq M$ and $F(x^k, y^k) \geq \underline{F}$ for any $k$, we have

$$c_k \Big( \varphi(x^k, y^k) - v_\gamma(x^k, y^k) \Big) \leq M - \underline{F}, \quad \forall k,$$

and we can obtain from $c_k = \underline{c}(k+1)^p$ that

$$\varphi(x^K, y^K) - v_\gamma(x^K, y^K) = O\left( \frac{1}{K^p} \right).$$

$\square$

## A.9 VERIFYING ASSUMPTION 3.2 FOR $g(x, y) = x\|y\|_1$

In this section, we prove that $g(x, y) = x\|y\|_1$ satisfies Assumption 3.2 when $X = \mathbb{R}_+$ and $Y = \mathbb{R}^m$.

Initially, as depicted in (Gao et al., 2023, Section 4.1), for $x \in \mathbb{R}_+$ and $y \in \mathbb{R}^m$,

$$x\|y\|_1 + \frac{\sqrt{p}}{2} x^2 + \frac{\sqrt{p}}{2} \|y\|^2 = \sum_{i=1}^{m} \frac{1}{2\sqrt{p}} \left( x + \sqrt{p}|y_i| \right)^2, \tag{60}$$

which is convex with respect to $(x, y) \in \mathbb{R}_+ \times \mathbb{R}^m$. Consequently, $g(x, y)$ is $\sqrt{p}$-weakly convex. Further, for any given $s \in (0, \bar{s}]$, we have

$$\text{Prox}_{s\tilde{g}(x, \cdot)}(\theta) = \text{Prox}_{sx\|\cdot\|_1}(\theta) = \mathcal{T}_{sx}(\theta) = (\mathcal{T}_{sx}(\theta_i))_{i=1}^m = ([|\theta_i| - sx]_+ \cdot \text{sgn}(\theta_i))_{i=1}^m. \tag{61}$$

This results in

$$\big\| \text{Prox}_{s\tilde{g}(x, \cdot)}(\theta) - \text{Prox}_{s\tilde{g}(x', \cdot)}(\theta) \big\| \leq s\|x - x'\| \leq \bar{s}\|x - x'\|. \tag{62}$$

In summary, Assumption 3.2 is satisfied by $g(x, y) = x\|y\|_1$ when $X = \mathbb{R}_+$ and $Y = \mathbb{R}^m$.

## A.10 MOREAU ENVELOPE BASED EFFICIENT FIRST-ORDER BILEVEL ALGORITHM (SMOOTH CASE)

---

**Algorithm 2** Moreau Envelope Based Efficient First-Order Bilevel Algorithm (Smooth Case)

---

**Initialize:** $x^0, y^0, \theta^0$, learning rates $\alpha_k, \beta_k, \eta_k$, proximal parameter $\gamma$, penalty parameter $c_k$;

1: **for** $k = 0, 1, \ldots, K - 1$ **do**
2:     update

$$\theta^{k+1} = \text{Proj}_Y \left( \theta^k - \eta_k \left( \nabla_y f(x^k, \theta^k) + \frac{1}{\gamma}(\theta^k - y^k) \right) \right),$$

$$x^{k+1} = \text{Proj}_X \left( x^k - \alpha_k \left( \frac{1}{c_k} \nabla_x F(x^k, y^k) + \nabla_x f(x^k, y^k) - \nabla_x f(x^k, \theta^{k+1}) \right) \right),$$

$$y^{k+1} = \text{Proj}_Y \left( y^k - \beta_k \left( \frac{1}{c_k} \nabla_y F(x^{k+1}, y^k) + \nabla_y f(x^{k+1}, y^k) - \frac{1}{\gamma}(y^k - \theta^{k+1}) \right) \right).$$

3: **end for**

---

For smooth BLO problems, specifically when $g(x, y) \equiv 0$ in (1), Algorithm 1 is specialized to Algorithm 2.

