# OpenReview forum: "Rethinking Moreau Envelope for Nonconvex Bi-Level Optimization: A Single-loop and Hessian-free Solution Strategy"
_ICLR.cc/2024/Conference — Submitted to ICLR 2024_

### Official Review · Reviewer_wVzv · 2023-10-19

**Soundness:** 2 fair
**Presentation:** 3 good
**Contribution:** 2 fair
**Rating:** 6
**Confidence:** 3

**Summary:**

This paper proposes a stochastic first-order algorithm based on the Moreau envelope reformulation.  Non-asymptotic convergence analysis under weaker conditions than previous works is provided. The proposed algorithm is evaluated on various setups, including few-shot learning, data hyper-cleaning, and neural architecture search.

**Strengths:**

1. The method looks novel.
2. Experiments look good.

**Weaknesses:**

1. Although the assumptions are indeed weaker than previous works, the convergence measure is also different from many previous works, so the results may not be directly comparable.
2. I think the recent paper which has a strong theoretical guarantee should be cited.
Kwon, Jeongyeol, et al. "On Penalty Methods for Nonconvex Bilevel Optimization and First-Order Stochastic Approximation." arXiv preprint arXiv:2309.01753 (2023).

**Questions:**

1. What does the "Rethinking" in the title mean? Why do we need to rethink? What is the finding after rethinking?

**Details Of Ethics Concerns:**

n/a.

---

> ### Author Response · Authors · 2023-11-15
>
> We sincerely appreciate the time and effort you dedicated to reviewing our work. Your constructive comments are much appreciated, and we would like to address each of the questions you raised below.
>
> (1) Although the assumptions are indeed weaker than previous works, the convergence measure is also different from many previous works, so the results may not be directly comparable.
>
> Reply:
> We appreciate your comment. In the realm of nonconvex-nonconvex BLO, we recognize that various works adopt distinct stationary measures. This diversity is acknowledged in our Table 1, stating, "Different methods utilize distinct stationary measures, precluding direct complexity comparisons."
>
> It is important to note that, traditionally, the hypergradient of the hyperobjective has been the preferred convergence measure in much of the existing literature. However, in our work, given the weaker assumptions we employ, the existence of the hypergradient is not always guaranteed.
>
> While acknowledging that comparing methods based on different stationary measures may lack full rigor, we wish to highlight our achievement in developing a non-asymptotic convergence analysis for our proposed method under these more lenient assumptions.
>
> While exploring the connection between various stationary measures for nonconvex-nonconvex BLOs is intriguing, it is not the primary focus of this work.
>
>
> (2) I think the recent paper which has a strong theoretical guarantee should be cited. Kwon, Jeongyeol, et al. "On Penalty Methods for Nonconvex Bilevel Optimization and First-Order Stochastic Approximation." arXiv preprint arXiv:2309.01753 (2023).
>
> Reply:
> Thank you for bringing this latest paper to our attention. We also noticed Kwon et al.'s paper after submitting our own. This paper primarily explores bilevel optimization from the perspective of the hyper-objective, using a penalty value function-based approach. It is relevant to our research and contributes to our understanding of nonconvex-nonconvex BLOs. We will cite it in our revised manuscript.
>
> (3) What does the "Rethinking" in the title mean? Why do we need to rethink? What is the finding after rethinking?
>
> Reply:
> The Moreau envelope, a widely recognized and frequently employed technique in single-level optimization problems, has only recently been adapted for bilevel optimization. This novel application was first introduced in the work of Gao et al. (2023) for convex lower-level objectives. In our current study, we expand upon this approach by extending the application from the convex setting, as explored by Gao et al. (2023), to a nonconvex context. Unlike Gao et al. (2023), which focused exclusively on a double-loop scheme, our work leverages the benefits of the Moreau envelope reformulation to propose a single-loop, Hessian-free scheme for BLO.

---

### Official Review · Reviewer_Qikw · 2023-10-22

**Soundness:** 2 fair
**Presentation:** 3 good
**Contribution:** 3 good
**Rating:** 5
**Confidence:** 4

**Summary:**

This paper concerns bi-level optimization (BLO) problems with an inner constraint set. By introducing the Moreau envelope of the lower-level function, the BLO can be reformulated into a nonconvex optimization problem with a smooth constraint. A single loop algorithm is proposed based on the formulation, leveraging the structure of the Moreau envelope. The author also provides a non-asymptotic rate for the algorithm and conducts numerical experiments to show its superiority.

**Strengths:**

1. The application of the Moreau envelope gives a nice reformation for the BLO. Compared with traditional value function reformation, the Moreau envelope-based reformation is a smooth problem and easier to solve.

2. Non-asymptotic convergence rate is developed for the proposed method without using the PL condition.

**Weaknesses:**

1. In the nonconvex case, (4) is only a relaxed version of the original BLO. Is not clear whether the global optimal solutions, local optimal solutions, and stationary points of (4) are related to the original BLO. Hence, I am not sure whether the stationarity measure in this paper is useful.

2. Assumption 3.2 (ii) appears too technical and artificial. The author does not convince me of its practicability, because the given examples are simple and no theoretical results is ensuring Assumption 3.2 (ii).

3. In Theorem 3.1, it is strange to say that "there exists c_{\alpha},c_{\beta}>0" as the upper bounds of the stepsizes $\alpha_k,\beta_k$. This appears to be impractical since only the existence of $c_{\alpha},c_{\beta}$ does not suggest how to choose the right stepsizes in implementation.

**Questions:**

Do the models in the experiments satisfy the Assumptions previously assumed?

**Details Of Ethics Concerns:**

This paper (ID 3145) is similar to submission 3552 of ICLR 2024 in many aspects, including the problem setting, idea, algorithm, theoretical analysis, and convergence results.

1.For the problem setting, both paper 3145 and paper 3552 focus on constrained Bilevel optimization. In the lower-level problem, paper 3145 considers a nonsmooth regularizer $g$ and paper 3552 considers constraints $g(x,y)\leq0$. The settings are very similar.

2.For the underlying ideas, the two papers both use the Moreau envelope to replace the value function in the lower-level problem. The difference is merely that paper 3552 applies the Lagrange duality due to the inequality constraints while paper 3145 does not.

3.For the algorithms, the updates of their algorithms are rather similar; see equations (9) (10) in paper 3145 and (10) (11) in paper 3552.

4.For the theoretical analysis, both papers use a merit function $V_k$ with similar structures and prove its sufficient decrease. Moreover, they consider similar stationarity measures and explain the stationarity measure via similar penalty functions; see equations (15) (16) in paper 3145 and (15) (16) in paper 3552.

5.For the convergence result, the obtained convergence rates $\frac1{K^{(1-2p)/2}}$, $\frac1{K^{p}}$, and $\frac1{K^{1/2}}$ are the same in the two papers; see Theorem 3.1 of paper 3145 and Theorem 3.1 of paper 3552.

I hope the authors can give some comments on the above similarities to address my concern.

---

> ### Author Response · Authors · 2023-11-15
>
> We appreciate the time you dedicated to reviewing our work and your constructive comments. Below, we would like to provide further clarification.
>
> (1) In the nonconvex case, (4) is only a relaxed version of the original BLO. Is not clear whether the global optimal solutions, local optimal solutions, and stationary points of (4) are related to the original BLO. Hence, I am not sure whether the stationarity measure in this paper is useful.
>
> Reply:
> We would like to provide further clarification:
>
> Firstly, the equivalence between the Moreau envelope reformulation (4) and the original BLO problem (1) is established in Section 2.1. Specifically, Theorem A.1 demonstrates that the reformulation (4) is equivalent to a specially relaxed version of the BLO problem (5), wherein the lower-level problem is replaced by its stationary points. The alignment of feasible sets in both (5) and (1) occurs when the lower-level problem's solution set aligns with its set of stationary points. Such conditions are met, for example, in scenarios where the lower-level objective is convex in $y$ or is smooth and adheres to the PL condition. Under these circumstances, the feasible regions defined by the reformulation (4) and the original BLO problem are identical, ensuring that their respective global and local optimal solutions concur.
>
> Regarding the stationarity of BLO problems, it is important to note that BLOs do not conform to standard nonlinear programming frameworks, and thus lack a universally accepted stationary condition. To address this, various equivalent reformulations associated with BLOs are explored for defining a suitable stationary condition. In this paper, we consider a novel reformulation (4). While the exploration of different stationary measures for BLOs is indeed a topic of interest, it falls outside the primary scope of our current research.
>
> (2) Assumption 3.2 (ii) appears too technical and artificial. The author does not convince me of its practicability, because the given examples are simple and no theoretical results is ensuring Assumption 3.2 (ii).
>
> Reply:
> We would like to provide additional clarification regarding Assumption 3.2(ii):
>
> Assumption 3.2(ii) is applicable to a broad spectrum of practical nonsmooth regularizers, many of which are yet to be explored in existing literature.
>
> This assumption is composed of three parts: the weak convexity of $g(x,y)$, the existence and Lipschitz continuity of $\nabla_x g(x,y)$, and the Lipschitz continuity of the proximal operator of $g(x,\cdot)$ with respect to the upper-level variable $x$, as specified in inequality (11). Notably, the first two components – weak convexity and the gradient's Lipschitz property – are generally mild and easily satisfied.
>
> Concerning the third component (condition (11)), when the nonsmooth component of the lower-level objective is independent of the UL variable, exemplified by $g(x,y)=\hat{g}(y)$, the condition is trivially met. As a result, Assumption 3.2(ii) is satisfied by standard convex regularizers such as $g(x,y)=\lambda \|y\|_1$ and $g(x,y)=\lambda \|y\|_2$, where $\lambda>0$. Furthermore, the assumption's requirement for only weak convexity in $g$ means it is also valid for common nonconvex regularizers like the Smoothly Clipped Absolute Deviation (SCAD) and the Minimax Concave Penalty (MCP).
>
> In scenarios where the nonsmooth component of the LL objective is dependent on the upper-level variable, condition (11) remains verifiable and practical. For instance, we studied the $l_1$ regularizer, $g(x,y)= x\| y\|_1$, with treating the regularization parameter as the upper-level variable, in Section A.9 of the Appendix, and demonstrated that $g(x,y)= x\| y\|_1$ meets Assumption 3.2(ii).
>
> In summary, Assumption 3.2(ii) is not only practical but also encompasses a wide range of nonsmooth regularizers not yet fully explored in previous research. The practical examples satisfying Assumption 3.2(ii) are discussed following Assumption 3.2 in our paper.
>
> (3) Do the models in the experiments satisfy the Assumptions previously assumed?
>
> Reply:
> In the smooth BLO scenario, our Assumptions simplify to both UL and LL objective functions being $L$-smooth. Consequently, it is straightforward to verify that the models in (17), (18), (20)-(23) satisfy the previously assumed Assumptions when both upper-level and lower-level variables are confined within a bounded set because both the UL and LL objectives are smooth.
>
> Turning to the non-smooth BLO scenario, specifically the LL Non-Smooth Case model (19) discussed in Section 4.1. In Appendix A.9, we establish that Assumption 3.2 holds for $g(x,y) = x \| y\|_1$. This verification can imply that the model in (19) satisfies the assumed Assumptions.
>
> Therefore, we can state that all models employed in our experimental studies meet the Assumptions assumed.

---

> ### Author Response · Authors · 2023-11-23
> **Response to Reviewer Qikw**
>
> We respectfully disagree with Reviewer Qikw's comments regarding the Ethics Concerns, which did not appear in the original version of this reviewer’s Official Review. It should be noted that, as indicated by the system time record, these newly added Ethics Concerns were included in the reviewer’s Official Review only after we had submitted our rebuttal to their initial review. Moreover, the reviewer did not respond to our rebuttal. Consequently, we did not receive any notifications from the system and only became aware of these newly added Ethics Concerns a few hours ago.
>
> Below, we provide detailed clarifications to address this biased comments.
>
>
>
> (1)   $\textbf{Differences in Problem Settings and Challenges:}$
>
> It's important to note that submissions 3145 and 3552 are tackling two distinct challenges that arise in the development of single-loop Hessian-free algorithms for bi-level optimization (BLO) problems. Specifically, the configurations of the BLO problems, and more specifically, the settings of the lower-level problems examined in submissions 3145 and 3552, are significantly different.
>
> To be more specific, submission 3145 focuses on BLO with an $\textbf{unconstrained}$ lower-level problem with potentially $\textbf{nonconvex}$ objective and a potentially additional $\textbf{nonsmooth}$ component depending on both upper- and lower-level variables and it is for addressing the nonconvexity and nonsmothness challenges appeared in the lower-level problem of BLO.
>
> In contrast, submission3552 addressed BLO problems with  $\textbf{constrained}$ convex lower-level problems.To provide more detail, the lower-level problem is with coupled lower-level constraints, and both the objective and constraint functions of lower-level problem is smooth and convex with respect to the lower-level variable. And submission3552 primarily addressed challenges arising from coupled lower-level constraints, i.e., lower-level $x$-dependent constraints.
>
>
> To summarize, the differences in problem settings and challenges can be outlined as follows.
> |                  | Submission3552 | Submission3145  |
> | :--------: | :--------: | :-------------: |
> |   $\textbf{Lower-level problem}$ | $\min_{y\in Y} f(x,y) \ \mathrm{s.t.}\  g(x,y)\leq 0$      |       $\min_{y\in Y} \varphi(x,y):=f(x,y)+g(x,y) $       |
> |   $\textbf{Annotation}$ | Here $g(x,y)$ is a coupled lower-level constraint      |       Here $g(x,y) $ is the nonsmooth part of the lower-level objective       |
> |   $\textbf{Challenges}$ | Lower-level $x$-dependent constraints, and non-differentiability in both value function and Moreau envelope-based reformulations      |       nonconvexity and nonsmoothness in LL objective      |

---

> > ### Author Response · Authors · 2023-11-23
> >
> > (2)$\textbf{Differences in Main Ideas and the Design of Algorithms:}$
> >
> > To address the differnet challenges encountered in BLO problems with unconstrained nonconvex nonsmooth lower-level problem and constrained convex lower-level problem, Submission 3145 and Submission 3552 developed different approaches and proposed distinct algorithms with using the same essential idea as in the value function-based reformulation, which is recently popularly applied and study in recent bilevel optimization works, see, e.g., Liu et al. (2021b; 2023a); Ye et al. (2022); Sow et al. (2022); Shen & Chen (2023); Kwon et al. (2023a); Lu & Mei (2023).
> >
> > Submission 3145, utilized $\textbf{Moreau envelope-based}$ reformulation of BLO, originally presented in Gao et al. (2023), combined with the idea of single-loop bilevel optimization in previous works, to design a single-loop and Hessian-free gradient-based algorithm. Thanks to the Moreau envelope-based reformulation and the strong convexity proximal term in the Moreau envelope, even when the convexity of lower-level problem or the PL condition usually required by other single-loop bilevel optimization works is absent, the error induced by the one step iteration of lower-level problem can controled. And adding a proximal operator of the nonsmooth component step in the iteration scheme also makes the propsed algorithm be able to handle the nonsmooth of the lower-level problem.
> >
> >
> >
> > In Submission 3552, inspired by the idea of proposing new value function to replace the value function in the value function approach reformulation in Gao et al. (2023),  a novel $\textbf{proximal Lagrangian value function}$ is introduced to handle constrained LL problem. By leveraging this function, a new single-level reformulation for constrained BLOs, converting them into equivalent single-level optimization problems with smooth constraints is proposed. The reason that Submission 3552 proposes a new proximal Lagrangian value function instead of uses Moreau envelope-based reformulation for handling constrained LL problem with coupled lower-level constraints is that as shown in Theorem 2.3 in Gao et al. (2023), the Moreau envelope of the constrained LL problem, as well as the value function of the constrained LL problem lacks differentiability if the solutions or the Lagrangian multipliers of the lower-level problem are not unique. However, the differentiability of the “value” function is very important for designing single-loop bilevel optimization algorithm, especially for controlling the error induced by the one step iteration of lower-level problem.
> >
> > To overcome this differentiability issue, with focusing on the convex LL problem scenario, Submission 3552 propose using the proximal Lagrangian value function, defined as the value function of a strongly-convex-strongly-concave min-max problem, exhibits the advantageous property of continuous differentiability, see Lemma A.1 within the Appendix of Submission3552. Since the problems and methods being addressed are different, the design of the algorithms also varies.

---

> > > ### Author Response · Authors · 2023-11-23
> > >
> > > (3)$\textbf{Differences in Theoretical Analysis:}$
> > >
> > > The proof frameworks for non-asymptotic convergence results in submissions 3145 (Theorem 3.1) and 3552 (Theorem 3.1) are similar to those of prior single-loop bilevel optimization works, as evidenced by their foundational approach to establishing a sufficient descent inequality for the merit function and subsequently deriving non-asymptotic estimations from such inequalities.
> > >
> > > To the best of our knowledge, this theoretical analysis approach for BLO originated from Ghadimi & Wang (2018), and has been widely employed in establishing the convergence rates of various algorithms, including stocBiO (Ji et al., 2021), ALSET (Chen et al., 2021), TTSA (Hong et al., 2023), AmIGO ( Arbel & Mairal, 2022a), FSLA (Li et al., 2022), SOBA and SABA (Dagréou et al., 2022), $F^2SA$ (Kwon et al., 2023a), V-PBGD (Shen & Chen, 2023) and others.
> > >
> > > Notwithstanding the structural similarity of the merit functions in submissions 3145 and 3552 to earlier works, these submissions introduce new elements due to the distinct problems, reformulations, and algorithms they encompass. Specifically, the hyper-objective components and the lower-level problem error terms, resulting from a one-step iteration strategy in prior works, are replaced by unique penalized objectives and corresponding iteration error terms in these submissions.
> > >
> > > (i) However, the different problem settings and reformulations in submissions 3145 and 3552 engender substantial differences in the techniques, arguments for the proof of a critical element—the sufficient descent inequality of the merit function—for their non-asymptotic convergence results, as elaborated in submissions 3145 (Lemma 3.1) and 3552 (Lemma 3.1). These disparities are primarily manifest in the proofs of auxiliary results.
> > >
> > > (ii) The methodologies and arguments employed in these auxiliary proofs differ markedly between the two submissions. In submission 3145, the proof of the descent lemma for $-v_\gamma(x,y)$ (Lemma A.3) hinges on the weak convexity of $v_\gamma(x,y)$, leveraging the properties of the Moreau envelope to obviate the need for convexity in the lower-level problem. Conversely, submission 3552 confronts a more complex scenario with its proximal Lagrangian value function $v_{\gamma,r}(x,y,z)$, defined as a min-max problem. The direct establishment of descent lemmas for $-v_{\gamma,r}(x,y,z)$ is notably challenging, prompting the derivation of variant descent lemmas for $-v_{\gamma,r}(x,y,\bar{z})$ and $-v_{\gamma,r}(\bar{x}, \bar{y}, z)$, with fixed $(\bar{x}, \bar{y})$ and $\bar{z}$ (Lemma A.2 and A.4). These variants require intricate techniques tailored to the min-max problem and also depend on the convexity of the lower-level problem.
> > >
> > > (iii) The demonstration of Lipschitz continuity for $\theta^*_\gamma(x,y)$ in Lemma A.5 of submission 3145 and for $(\theta^*(x,y,z), \lambda^*(x,y,z))$ in Lemma A.3 of submission 3552 also exhibit pronounced differences. In submission 3145, addressing the non-smooth term in the lower-level problem necessitates employing a fixed-point characterization of $\theta^*_\gamma(x,y)$. This is combined with leveraging the Lipschitz property with respect to $x$ of the proximal operator for the non-smooth component $g(x,y)$ to establish the Lipschitz continuity of $\theta^*_\gamma(x,y)$. Conversely, in submission 3552, the pair $(\theta^*(x,y,z), \lambda^*(x,y,z))$ represents a saddle point in a min-max problem, necessitating a recharacterization of this saddle point as a solution to a generalized equation. Subsequently, the application of monotone operator theorem results is crucial to proving their Lipschitz continuity.
> > >
> > > (iv) Regarding Lemma A.6 in submission 3145 and Lemma A.3 in submission 3552, the contrast in the generation of $\theta^{k+1}$ through a proximal gradient step in a single-level optimization problem, and $(\theta^{k+1}, \lambda^{k+1})$ through a forward-backward step in a min-max problem, necessitates distinct approaches in their contraction property proofs.
> > >
> > > $\textbf{In summary}$, while submissions 3145 and 3552 adhere to the overarching proof structures of non-asymptotic convergence results characteristic of prior single-loop bilevel optimization studies, they necessitate significantly varied and specialized techniques. These adaptations address specific challenges and complexities arising from non-convexity and non-smoothness in the lower-level problem of the bilevel programs considered in submission 3145, and the lower-level coupled constraints in the bilevel programs considered  in submission 3552, which are pivotal in proving non-asymptotic convergence.

---

> > > > ### Author Response · Authors · 2023-11-23
> > > >
> > > > (4)$\textbf{Differences in Experiments:}$
> > > >
> > > > Submission3145 validate the effectiveness and efficiency of the proposed algorithm MEHA on various bilevel optimization problems with nonconvex lower-level problem, such as, few-shot learning, data hyper-cleaning and the real-world neural architecture search application.
> > > >
> > > >
> > > > Submission3552 evaluate the efficiency of the proposed algorithm LV-HBA through numerical experiments on various bilevel optimization problems with lower-level problem with couples constraints, such as synthetic problems, hyperparameter optimization for SVM and federated bilevel learning.

---

### Official Review · Reviewer_y7ue · 2023-10-28

**Soundness:** 3 good
**Presentation:** 3 good
**Contribution:** 4 excellent
**Rating:** 8
**Confidence:** 5

**Summary:**

This paper studied the nonconvex constrained bilevel optimization, where its upper level is nonconvex and its lower level is nonsmooth and weakly convex. It proposed an efficient single-loop gradient-based Hessian-free algorithm based on the Moreau envelope technique. Moreover, it provided the non-asymptotic convergence analysis for the proposed algorithm. Extensive experimental results demonstrate the efficiency of the proposed algorithm. In summary, the contributions of this paper are significant on the design method and solid convergence analysis.

**Strengths:**

This paper proposed an efficient single-loop gradient-based Hessian-free algorithm based on the Moreau envelope technique. Moreover, it provided the non-asymptotic convergence analysis for the proposed algorithm. Extensive experimental results demonstrate the efficiency of the proposed algorithm. In summary, the contributions of this paper are significant on the design method and solid convergence analysis.

**Weaknesses:**

It is better to list some bilevel optimization examples in machine learning that have non-smooth and weakly convex lower level functions, which will strength the motivation of this work.

**Questions:**

Some comments:

1)	In the proposed algorithm 1, there exist five tuning parameters $\alpha_k,\beta_k,\eta_k,\gamma,c_k$. Although the authors gave the range of these parameters in the convergence analysis, I still think the choice of these tuning parameters is not easy in practice.

2)	From the convergence analysis, I saw that the authors used the condition that $f(x,y)$ is a weakly convex. I suggest that the authors should add this condition in Assumption 3.2 of the paper.

3)	The inequality (11) in Assumption 3.2 (ii) is a strict condition ? If not , please give an example.

4)	It is better to list some bilevel optimization examples in machine learning that have nonsmooth and weakly convex lower level functions, which will strength the motivation of this work.

---

> ### Author Response · Authors · 2023-11-15
>
> We sincerely appreciate the time and effort you dedicated to reviewing our work and are thankful for your constructive comments. In the following section, we will address each question.
>
> (1) It is better to list some bilevel optimization examples in machine learning that have non-smooth and weakly convex lower level functions, which will strength the motivation of this work.
>
> Reply:
> Thank you for your valuable suggestions. Non-smooth and weakly convex functions, like regularization (e.g., $\ell_1$), find extensive use in various machine learning applications. These applications include sparse representation, sparse coding, feature selection, and hyper-parameter optimization. For instance, in the context of hyper-parameter optimization for low-level vision tasks, $\ell_1$ regularization is applied to enforce sparsity in image restoration. Additionally, in data hyper-cleaning, this same regularization technique can help mitigate overfitting in the learning model.
>
> In Appendix A.1, we provide a brief review of recent works on nonsmooth BLO. We appreciate your feedback and plan to incorporate more bilevel optimization examples in the introduction section of the revised manuscript.
>
> (2) In the proposed algorithm 1, there exist five tuning parameters $\alpha_k$, $\beta_k$, $\eta_k$, $\gamma$, $c_k$. Although the authors gave the range of these parameters in the convergence analysis, I still think the choice of these tuning parameters is not easy in practice.
>
> Reply:
> To explore the sensitivity of these parameters, we conducted extra numerical experiments in the LL Non-Smooth Case, with a dimension of 1000. The table below displays convergence time and steps while altering one parameter, keeping the others constant. It's important to note that the algorithm achieves convergence with various parameter combinations.
>
> | | | | | | | | |
> |-|-|-|-|-|-|-|-|
> |Strategy|$\alpha$|$\beta$|$\eta$|$\gamma$|$\underline{c}$|Steps|Time (s)|
> |Original|0.1|0.00001|0.1|10|2|97|22.83|
> |Changing $\alpha$|0.05|0.00001|0.1|10|2|109|25.55|
> |Changing $\alpha$|0.5|0.00001|0.1|10|2|88|13.66|
> |Changing $\beta$|0.1|0.00002|0.1|10|2|91|22.01|
> |Changing $\beta$|0.1|0.00003|0.1|10|2|85|13.41|
> |Changing $\eta$|0.1|0.00001|0.5|10|2|88|12.82|
> |Changing $\eta$|0.1|0.00001|0.01|10|2|199|51.99|
> |Changing $\gamma$|0.1|0.00001|0.1|2|2|89|19.19|
> |Changing $\gamma$|0.1|0.00001|0.1|100|2|90|15.3|
> |Changing $\underline{c}$|0.1|0.00001|0.1|10|10|105|17.52|
> |Changing $\underline{c}$|0.1|0.00001|0.1|10|50|107|17.22|
>
> (3) From the convergence analysis, I saw that the authors used the condition that $f(x,y)$ is a weakly convex. I suggest that the authors should add this condition in Assumption 3.2 of the paper.
>
> Reply:
> We appreciate your suggestion. As discussed in the paragraph prior to Section 3.2, the application of the descent lemma allows us to deduce that a function possessing a Lipschitz-continuous gradient is weakly convex. Consequently, Assumption 3.2(i) is sufficient for the weak convexity of the function $f(x, y)$.
>
> (4) The inequality (11) in Assumption 3.2 (ii) is a strict condition ? If not , please give an example.
>
> Reply:
> We would like to clarify that Assumption 3.2(ii) concerning the nonsmooth component $g(x,y)$ is not restrictive and is applicable to a wide range of practical nonsmooth regularizers, many of which have not been explored in prior research.
>
> In our paper, under Assumption 3.2, we discuss practical instances that fulfill Assumption 3.2(ii). Notably, when the nonsmooth component of the lower-level objective is independent of the upper-level variable, exemplified as $g(x,y)=\hat{g}(y)$, Assumption 3.2(ii) is satisfied for convex regularizers like $g(x,y)=\lambda|y|_1$ and $g(x,y)=\lambda|y|_2$ where $\lambda>0$. Additionally, even for typical nonconvex regularizers such as the Smoothly Clipped Absolute Deviation (SCAD) and the Minimax Concave Penalty (MCP), Assumption 3.2(ii) remains valid.
>
> Furthermore, in cases where the nonsmooth component of the lower-level objective is dependent on the upper-level variable, we investigate a specific example: $g(x,y)= x\|y\|_1$. Here, $g(x,y)$ represents the $l_1$ regularizer, considering the regularization parameter as the upper-level variable. As demonstrated in Section A.9 of the Appendix, $g(x,y)= x\|y\|_1$ also adheres to Assumption 3.2(ii).
>
> Thus, we assert that Assumption 3.2(ii) is a practical and non-restrictive condition, encompassing a broad spectrum of practical scenarios.
>
> (5) It is better to list some bilevel optimization examples in machine learning that have nonsmooth and weakly convex lower level functions, which will strength the motivation of this work.
>
> Reply:
> Thank you for your suggestion. Please refer to our previous response for addressing your concern.

---

> > ### Comment · Reviewer_y7ue · 2023-11-23
> > **Reply to Authors**
> >
> > Thanks for your responses. My concerns have been well-addressed, so I keep my score.

---

### Official Review · Reviewer_am3o · 2023-10-29

**Soundness:** 3 good
**Presentation:** 3 good
**Contribution:** 3 good
**Rating:** 6
**Confidence:** 4

**Summary:**

The paper applies the Moreau envelope for solving bilevel optimization with non-convex low-level optimization. The proposed algorithm extends such Moreau envelope framework from convex to non-convex settings, achieves a single loop structure, and avoids Hessian computation at the same time. A non-asymptotic convergence rate is derived and extensive numerical evaluations demonstrate faster convergence and superior performance.

**Strengths:**

1. Compared to previous bilevel optimization algorithms, the proposed MEHA algorithm is both single-loop and hessian-free, yielding an advantage in the efficiency of convergence.

2. The numerical evaluation conducted is quite comprehensive, covering both synthetic experiments and various real-world tasks.

**Weaknesses:**

The only weakness the reviewer sees is the lack of technical novelty, as the analysis is largely based on previous work utilizing the Moreau envelope for convex low-level objectives (Gao et al., 2023), and other Hessian-free works. As a result, the proposed method seems like an extension / combination of previous methods.

**Questions:**

N/A

---

> ### Author Response · Authors · 2023-11-15
>
> Thank you for taking the time to read and review our work. We appreciate your feedback and would like to provide further clarification.
>
> (1) The only weakness the reviewer sees is the lack of technical novelty, as the analysis is largely based on previous work utilizing the Moreau envelope for convex low-level objectives (Gao et al., 2023), and other Hessian-free works. As a result, the proposed method seems like an extension / combination of previous methods.
>
> Reply:
> Our research indeed builds upon existing works, such as the use of the Moreau envelope for convex lower-level objectives (Gao et al., 2023) and other Hessian-free works. However, our contribution goes beyond merely synthesizing these existing approaches, as we introduce several novel elements.
>
> Firstly, while the Moreau envelope reformulation was initially proposed in Gao et al. (2023) with proposing a corresponding double-loop scheme, our work extends this to a non-convex setting and propose a new single-loop algorithm. Our single-loop algorithm's convergence analysis necessitated proving the Lipschitz continuity of the Moreau envelope solution $\theta_{\gamma}^*(x,y) :=  \mathrm{argmin}_{\theta \in Y} \{ \varphi(x, \theta) + \frac{1}{2\gamma}\|\theta-y\|^2 \}$, particularly challenging in the presence of a nonsmooth component, $g(x,y)$, in the lower-level objective (2). This analysis, detailed in Lemma A.5, represents a significant technical advancement.
>
> Furthermore, while previous Hessian-free approaches for BLO with non-convex lower-level problems, such as those by Ye et al. (2022) and Shen & Chen (2023), depended on the Polyak-Łojasiewicz (PL) condition, our method does not. The Moreau envelope technique does not inherently satisfy the PL condition, prompting us to develop new techniques distinct from previous Hessian-free methods.
>
> Moreover, to the best of our knowledge, no existing Hessian-free algorithms have addressed scenarios where the lower-level problem objective includes a nonsmooth component, as investigated in our work. Our iteration scheme and convergence analysis for managing this nonsmooth component are thus unique contributions to the field of Hessian-free algorithm research.
>
> In summary, our research extends the Moreau envelope-based algorithm from Gao et al. (2023) from a double-loop to a single-loop approach. Additionally, we are the first to explore a Hessian-free algorithm for bilevel optimization problems where the lower-level objective incorporates a nonsmooth component, all without relying on the PL condition.

---

> > ### Comment · Reviewer_am3o · 2023-11-22
> >
> > Thanks for the response. My concerns have been well-addressed and I'll keep my score and increase my confidence.

---

### Official Review · Reviewer_UhJL · 2023-10-30

**Soundness:** 3 good
**Presentation:** 3 good
**Contribution:** 4 excellent
**Rating:** 8
**Confidence:** 2

**Summary:**

The authors propose a single-loop algorithm for bilevel optimization problems.
The algorithm can handle non-smooth and smooth + non-smooth weakly convex inner functions. The authors provide convergence rates under very weak assumptions (smoothness of the outer and the inner problem).

**Strengths:**

To my knowledge, this work contains two major novelties:
- in the smooth case, convergence proof a single loop algorithm with weak assumptions (smoothness of the inner and the outer problem only, no need for bounded gradients)
- in the non-smooth case, to my knowledge, this is the first single-loop algorithm proposed
Maybe the authors and other reviewers can comment on this.

**Weaknesses:**

While the proposed algorithm is clearly defined, authors could do a better job at providing the intuitions: the directions $d_x^k$ and $d_y^k$ are given with little context. The same comment applies to the merit function and Lemma 3.1.

**Questions:**

- Could you comment on the novelty, is this the first analysis with such a weak set of assumptions? Or I am missing something?

- Could you give intuitions on the proof? (which is currently 10 pages long in the appendix) and intuitions on the merit function.
Maybe the authors could provide a proof sketch

---

> ### Author Response · Authors · 2023-11-15
>
> We are grateful for your dedicated time in reviewing our work and your constructive comments. Below, we address each question.
>
> (1) While the proposed algorithm is clearly defined, authors could do a better job at providing the intuitions: the directions $d_x^k$ and $d_y^k$ are given with little context. The same comment applies to the merit function and Lemma 3.1.
>
> Reply:
> Thank you for the suggestions. In the initial manuscript, we elucidated that the update scheme for variables $(x, y)$ can be interpreted as an inexact alternating proximal gradient approach, applied to a nonsmooth optimization problem:
> \begin{equation*}
> \min_{(x,y)\in X \times Y} \frac{1}{ c_k}F(x,y) + f(x, y) + g(x,y) - v_\gamma (x,y).
> \end{equation*}
> Specifically, $d_{x}^k$ approximates the gradient of $\frac{1}{ c_k}F(x,y) + f(x, y) + g(x,y) - v_\gamma (x,y)$ with respect to $x$ at point $(x^k, y^k)$, with $\theta^{k+1}$ as an approximation for $\theta^*_{\gamma}(x^k,y^k) $. Similarly, $d_{y}^k$ approximates the gradient of $\frac{1}{ c_k}F(x,y) + f(x, y) - v_\gamma (x,y)$ with respect to $y$ at $(x^{k+1}, y^k)$, using $\theta^{k+1}$ to approximate $\theta^*_{\gamma}(x^{k+1},y^k) $.
>
> Regarding the merit function $V_k$, employed in convergence analysis and Lemma 3.1, it is defined as the scalar sum of the penalized problem’s objective (12):
> \begin{equation*}
> \phi_{c_k}(x,y) := \frac{1}{ c_k}(F(x,y) - \underline{F}) + f(x, y) + g(x,y) - v_\gamma (x,y),
> \end{equation*}
> and the proximity of iterate $\theta^k$ to the actual solution defining the Moreau envelope of lower-level problem at iterate $(x^k, y^k)$, i.e.,
> $\mathrm{argmin}_{\theta\in Y} \{\varphi(x^k,\theta)+\frac{1}{2\gamma}\|\theta-y^k\|^2 \}$.
>
> (2) Could you comment on the novelty, is this the first analysis with such a weak set of assumptions? Or I am missing something?
>
> Reply:
> In this study, we present a novel algorithm for BLO by leveraging the Moreau envelope-based reformulation initially introduced by Gao et al. (2023). To our knowledge, this is the first application of such a reformulation in the design of a single-loop, Hessian-free gradient-based algorithm for BLO with such weak assumptions. Significantly, our algorithm has a non-asymptotic convergence analysis, and can handle lower-level problems with potentially nonconvex and nonsmooth objectives. Notably, this result is achieved without the necessity of the PL condition on the lower-level problem.

---

> ### Author Response · Authors · 2023-11-15
>
> (3) Could you give intuitions on the proof? (which is currently 10 pages long in the appendix) and intuitions on the merit function. Maybe the authors could provide a proof sketch
>
> Reply:
> Intuitions on the merit function $V_k$:
>
> The update scheme for variables $(x, y)$ can be interpreted as an inexact alternating proximal gradient method applied to the penalized problem (12) :
> \begin{equation*}
> \phi_{c_k}(x, y) := \frac{1}{c_k}\big(F(x, y) - \underline{F} \big) + f(x, y) + g(x, y) - v_\gamma(x, y).
> \end{equation*}
> Consequently, we anticipate a decreasing value of $\phi_{c_k}(x^k, y^k)$ for the iterates $(x^k, y^k)$.
>
> Furthermore, the update of variable $\theta$ aligns precisely with a single proximal gradient step to the problem defining the Moreau envelope $v_{\gamma}(x,y) :=\inf_{\theta \in Y} \{ \varphi(x, \theta) + \frac{1}{2\gamma}\|\theta-y\|^2 \}$. This alignment leads us to expect that the iterate $\theta^k$ will converge towards the solution of the Moreau envelope at $(x^k, y^k)$, specifically $ \mathrm{argmin}_{\theta \in Y} \{ \varphi(x^k, \theta) + \frac{1}{2\gamma}|\theta - y^k|^2 \}$.
>
> Building on these insights into the update schemes for $(x, y)$ and $\theta$, we examine the merit function $V_k$ defined as the sum of the objective value of the penalized problem (12), $\phi_{c_k}(x, y)$, and the distance between the iterate $\theta^k$ and the optimal solution $\theta_{\gamma}^*(x^k, y^k)$ for the Moreau envelope of lower-level problem at $(x^k, y^k)$.
>
> Proof Sketch of non-asymptotic convergence result:
>
> The proof of Theorem 3.1, which establishes the non-asymptotic convergence, fundamentally relies on the monotonically decreasing property of the merit function $V_k$, as delineated in Lemma 3.1. We succinctly outline the pivotal steps leading to this lemma.
>
> Step 1: We first consider the Moreau envelope $v_{\gamma}(x,y)$, focusing on two of its critical properties: its weak convexity (referenced in Lemma A.1) and the associated gradient formulas (outlined in Lemma A.2). These properties enable us to derive an upper bound for the descent of the objective value of the penalized problem (12), $\phi_{c_k}(x, y)$ with incorporating the error term $\left(\frac{\alpha_k}{2} (L_f + L_g)^2 + \frac{\beta_k}{\gamma^2} \right) \left\| \theta^{k+1} - \theta_{\gamma}^*(x^k,y^{k}) \right\|^2$, as formulated in Equation (35) in Lemma A.7.
>
> Step 2: The subsequent step involves leveraging the Lipschitz continuity of the Moreau envelope solution $\theta_{\gamma}^*(x,y)$ (as per Lemma A.5) along with the contraction properties of $\theta^k$ towards this solution (discussed in Lemma A.6). This approach is instrumental in controlling the aforementioned error term $\left\| \theta^{k+1} - \theta_{\gamma}^*(x^k,y^{k}) \right\|^2$ as presented in Equation (35).
>
> Ultimately, these steps culminate in confirming the monotonically decreasing nature of the merit function $V_k$ as in Lemma 3.1.

---

### Meta-Review · Area_Chair_9YbH · 2024-01-15

**Metareview:**

The paper considers a class of bilevel optimization problems, in which the upper-level objective has Lipschitz gradient, the lower-level involves an objective of a composite form, and the variables are constrained to be in closed convex sets. It proposes a Moreau envelope-based reformulation and then develops a single-loop first-order algorithm for solving it. The paper then provides a non-asymptotic convergence analysis and numerical results of the proposed algorithm.

The PCs have taken up the discussion of the paper due to possible overlap with another ICLR submission (3552). The authors are advised to cite related works (including those by themselves) and clearly delineate the differences in the approaches and results in the submissions. On the technical side, while the authors responded to the reviewers' comments on the strength of the assumptions, it remains unclear whether the real-world instances tested in the experiments satisfy the proposed assumptions. Moreover, the incomparability of the convergence measure used in this paper with those in the literature somewhat weakens the technical contributions. The paper can benefit from a substantial revision before publishing.

**Justification For Why Not Higher Score:**

The technical contributions are limited due to the issues mentioned above.

**Justification For Why Not Lower Score:**

N/A

---

### Decision · Program_Chairs · 2024-01-16

Reject